



# Characterizing atmospheric stability in complex terrain

Nathan J. Agarwal[1] and Julie K. Lundquist[1,2]

[1]Department of Earth and Planetary Sciences, Johns Hopkins University, Baltimore, MD, USA
[2]National Renewable Energy Laboratory, Golden, CO, USA

**Correspondence:** Nathan J. Agarwal (nagarw22@jh.edu)

**Abstract.** Characterizing atmospheric stability becomes challenging in heterogeneous complex terrain. We use data from 47 meteorological towers associated with the Perdigão field campaign to recommend data processing approaches and to assess the limitations of shorter or fewer towers. We quantify atmospheric stability according to the Obukhov length, the turbulence kinetic energy, and the turbulence dissipation rate using a range of decomposition periods including consistent 10 minute periods to match convention in the wind energy community and consistent 30 minute periods to match convention in the atmospheric science community. Atmospheric stability characterization is impacted by the Reynolds decomposition period, so care should be taken to use appropriate intervals. Additionally, 10 m measurements do not provide reliable 100 m hub-height stability predictions. Finally, we demonstrate a methodology that can indicate the necessary number and location of towers to characterize atmospheric stability. Holistically, this work addresses challenges in relying on sparse surface measurements.

*Copyright statement.* This work was authored in part by the National Renewable Energy Laboratory for the U.S. Department of Energy (DOE) under contract no. DE-AC36-08GO28308. Funding was provided by the U.S. Department of Energy Office of Energy Efficiency and Renewable Energy Wind Energy Technologies Office. The views expressed in the article do not necessarily represent the views of the DOE or the U.S. Government. The U.S. Government retains and the publisher, by accepting the article for publication, acknowledges that the U.S. Government retains a nonexclusive, paid-up, irrevocable, worldwide license to publish or reproduce the published form of this work, or allow others to do so, for U.S. Government purposes.

## 1 Introduction

Atmospheric stability describes how the atmosphere responds to disturbances. Static stability considers how buoyancy may affect a vertically perturbed air parcel. A region of the atmosphere is statically stable if a perturbed parcel in that region is restored back toward its equilibrium height, statically unstable if the air parcel rises away from its equilibrium height, and neutral if the air parcel is unaffected. Dynamic stability considers the effects of both buoyancy and wind shear processes to determine whether a flow will become turbulent. Statically stable flows can be dynamically unstable and can become turbulent if the wind shear is strong enough. Thus, laminar flow exists only if the layer of air is stable both dynamically and statically (Stull, 1988; Angevine et al., 2020). Because both static and dynamic stability address key characteristics of the wind resource, atmospheric stability characterization remains an active field of study.





Atmospheric stability affects wind power generation. Optis and Perr-Sauer (2019) implemented several machine-learning methods to assess the relative importance of meteorological variables in wind farm energy assessments and found atmospheric stability to be one of the most important variables of those considered, regardless of the model used. Stability-driven differences in power generation have been documented for both onshore and offshore projects and may affect turbines differently, depending on the turbine's relative location. Wharton and Lundquist (2012) analyzed turbine power generation data for a west coast North American wind farm in channeled flow and found average power generation differences between stable and unstable cases approaching 15%. Pérez et al. (2023) analyzed meteorological mast and marine buoy data for a Yucatan coastal site and found 17% difference in power output between stable and unstable conditions. Radünz et al. (2021) investigated two wind farms in the complex Morrinhos site in Northeast Brazil and found that while the front rows outperformed the back rows during daytime unstable conditions, the back rows outperformed the front rows during the nighttime very stable conditions.

Atmospheric stability also affects wind power generation through the effects of turbulence intensity, shear, and veer (Lundquist, 2022). In very early work, Elliott (1990) analyzed power curves for three turbines and showed that significant errors in power curve measurements can result if the stability-influenced effects of wind shear and turbulence are ignored. Later, Vanderwende and Lundquist (2012) analyzed nacelle and meteorological tower measurements from a wind farm in the high plains of central North America and found that stable conditions were associated with reduced power performance of up to 6%. Similarly, Bardal et al. (2015) analyzed lidar and meteorological mast data for a central Norwegian coastal wind farm and found reduced power in the middle of the power curve for cases with shear exponents greater than 0.15. Gao et al. (2021) collected five years of data at the Eolos facility in Minnesota and also found that the shape of the wind veer profile affects differences in power performance. Sanchez Gomez and Lundquist (2020) studied power production at an onshore wind power plant with a strong diurnal cycle and found that while large values of wind veer were associated with turbine underperformance, large shear values with small values of wind veer were associated with turbine overperformance, resolving the apparent contradiction between Wharton and Lundquist (2012) and Vanderwende and Lundquist (2012).

The stability-driven effects of turbulence, wind shear, and wind veer also affect turbine loads. The aeroelastic simulations performed in Sathe et al. (2013) showed that rotor and tower loads increase in high shear environments. Further, in Dimitrov et al. (2018), while wind shear was shown to strongly influence blade root loads in surrogate load models calibrated with a database of high-fidelity load simulations, veer showed only a minimal effect on fatigue loads. Wind shear also dominated the sensitivity analysis performed in Robertson et al. (2019) with 18 different parameters influencing blade root out-of-plane pitching moment and blade shaft bending moments. Putri et al. (2019) found that the fatigue damage of a spar-buoy offshore wind turbine is higher in unstable cases than in near-neutral cases.

Atmospheric stability also affects turbine wakes. Baker and Walker (1984) observed a slower wake recovery for a wind turbine under stable conditions, and Iungo and Porté-Agel (2014) showed a faster wake recovery under convective conditions. Magnusson and Smedman (1994) showed a larger velocity deficit in stable conditions for a turbine near the Alsvik wind farm. The scanning lidar measurements of Aitken et al. (2014) also found slightly stronger wake deficits in stable conditions even for a turbulent site, while the scanning lidar measurements of Bodini et al. (2017) showed that wakes are much more distinct in stable conditions. Wind turbine wakes also respond to the stability-driven effects of wind veer. Large-eddy simulation studies





(Vollmer et al., 2016; Churchfield and Sirnivas, 2018), scanning lidar measurements (Bodini et al., 2017), and nacelle lidar measurements (Brugger et al., 2019) have shown that in the veering flow typical of stably stratified conditions, the wake will stretch into an ellipse. Englberger and Dörnbrack (2018) performed a detailed parameter large-eddy simulation study that showed that increasing veer increases the erosion of the turbine wake. Stability-based effects on wake propagation also have secondary effects on turbine power generation (Barthelmie et al., 2013). Keck et al. (2014) compared the results of their

dynamic wake meandering model to field observations and showed 22% differences in wake losses between stable and unstable conditions.

Given that atmospheric stability has implications for power generation, turbine loads, and wake propagation, determining optimal atmospheric stability definitions is critical for reliable wind resource assessments and wind energy forecasting (Optis and Perr-Sauer, 2019; Lee et al., 2020). However, multiple metrics exist to quantify stability. Some metrics, like the Obukhov

length, classify stability at a single altitude. Other metrics, like the bulk and gradient Richardson numbers, classify stability across a given vertical extent. Other metrics still, like the turbulence kinetic energy ($TKE$) and turbulence dissipation rate ($\epsilon$), characterize stability indirectly through turbulence (Stull, 1988). Some metrics also rely on determining an appropriate averaging window, and there are multiple – sometimes conflicting – approaches for defining the averaging window (Howell and Mahrt, 1994).

Complex terrain introduces further challenges to atmospheric stability classification (Serafin et al., 2018; Cantero et al., 2022; Mosso et al., 2024, 2025). Skillful wind forecasts in complex terrain are challenging, owing largely to the prevalence of heterogeneous terrain-modulated flows, such as mountain wakes, mountain waves, gap flows, valley cold pools, and mountain-valley circulations (Fernando et al., 2019). Despite these issues, many modeling efforts still rely on physical parameterizations developed over flat, horizontally homogeneous terrain because of constrained data availability (Stiperski and Rotach, 2016;

Sfyri et al., 2018).

These complex terrain dynamics influence wind energy siting and further complicate wind energy assessments (Clifton et al., 2022). Lange et al. (2017) used a scale model in a three-dimensional wind testing chamber and showed that the mean wind, wind shear, and turbulence level are extremely sensitive to the exact details of the terrain, affecting the lifetime and maintenance costs of wind turbines. Han et al. (2018) analyzed experimental data from a wind farm site in complex terrain in China and

found that complex terrain modifies the wind profiles, resulting in significant performance differences under both stable and unstable conditions. Despite their impact, these dynamics are still not captured well by commonly used commercial wind farm planning computer models (Shaw et al., 2019; Olson et al., 2019).

Financial constraints also inform challenges to atmospheric stability characterization. While meteorological towers can provide both the wind and temperature measurements required to assess atmospheric stability, the material costs for meteoro-

logical towers alone can cost on the order of $1 million for a 100 MW utility-scale onshore project (Eberle et al., 2019). Sonic anemometers to measure wind characteristics and thermometers to measure temperatures introduce additional costs. Thus, it could be possible to reduce costs if surface stability assessments were more reliable in predicting hub-height stability without the need to invest in tall (i.e., 100 m) meteorological towers.





This work addresses the representativeness of stability metrics by assessing three stability metrics based on their ability to capture changes in time, height, and terrain. Based on the results from this analysis, this work offers guidance to meteorologists and other stakeholders that may be interested in analyzing complex terrain sites. This manuscript is organized as follows. The Perdigão field campaign is introduced, the data preparation is described, and both the metrics and evaluation criteria are outlined in Sect. 2. Then, the results from these assessments are presented in Sect. 3 and discussed in Sect. 4. Finally, conclusions are drawn and recommendations are offered in Sect. 5.

## 2 Data and methods

### 2.1 Perdigão field campaign

The Perdigão field campaign, which occurred 1 May–15 June 2017, was a scientific research effort near the town of Perdigão in central Portugal. This international collaboration addressed knowledge gaps surrounding the microscale details of winds in complex terrain by collecting a comprehensive, high-resolution dataset of meteorological quantities for the site (Fernando et al., 2019). The field campaign took place in the Vale do Cobrão, with an approximately two-dimensional valley and parallel ridges with annual 10 m wind climatology perpendicular to the ridges (Fig. 1).

Multiple instruments were deployed throughout the field campaign. Instrumentation included 49 meteorological towers, radiosondes, lidars, two tethered lifting systems, and additional instrumentation including radars and radiometers (Fernando et al., 2019).

The meteorological towers provide the data for this analysis. Of the 49 towers, 47 included a 20 Hz sonic anemometer at the 10 m ("surface") level. Of these 47 towers, 18 also included a 1 Hz thermometer at this 10 m level, although the only 10 m temperature measurements that were used for this analysis were from the three 100 m towers. All towers were constructed to minimize flow distortion by the tower. In cases where local obstacles interfered with the anemometer location, the anemometer boom was set to point down-valley. Booms on the towers pointed toward directions varying between 111 and 166 degrees. The azimuth, pitch, roll, and height of the sonic anemometer mounting booms were measured and recorded. The sonic anemometer data were then tilt-corrected using the double-rotation method following the approach of Wilczak et al. (2001).

The three 100 m towers were strategically placed to sample flow conditions within differing terrains. Specifically, tse04 was located on an exposed ridge on the southwest, tse09 in the valley with eucalyptus and fir trees, and tse13 on a ridge with a heterogeneous canopy to the northeast (Table 1). Although translating CORINE Land Cover data (Bossard et al., 2000) into U.S. Geological Survey land use types to obtain surface roughness lengths (Pineda et al., 2004) suggests surface roughness lengths on the order of 0.01–0.05 m, Wise et al. (2021), Wagner et al. (2019), and Palma et al. (2020) have suggested that larger surface roughness lengths are more appropriate for the site. For each 100 m tower, 20 Hz sonic anemometers and 1 Hz temperature sensor measurements were collected at 10, 20, 40, 60, 80, and 100 m (Fernando et al., 2019; EOL, 2019), among other measurements. The 60 m temperature measurements for tse13 (NE ridge) were not used here due to their high (76%) data unavailability.



**Figure 1.** Topographic map of the Perdigão valley field campaign site with meteorological towers included. Extensive other instrumentation are not depicted.

**Table 1.** Shortname Reference for 100 m towers

| Tower | Shortname | Topography | Location [WGS84] | Elevation above sea level [m] |
|-------|-----------|-----------|------------------|-------------------------------|
| tse04 | Southwest Ridge | Turbine Ridge | $7°44'33.37''$W $39°42'21.47''$N | 473 |
| tse09 | Valley | Valley | $7°44'5.40''$W $39°42'40.36.''$N | 305 |
| tse13 | Northeast Ridge | Canopy Ridge | $7°43'49.38''$W $39°42'48.97.''$N | 453 |





## 2.2 Data preparation

The analysis presented herein relies on the 20 Hz sonic anemometer and 1 Hz air temperature sensor measurements from the meteorological towers. These tilt-corrected data were also screened for quality assurance according to standard sensor error flags, with the most prevalent flag being the effect of rain (EOL, 2019). These data were also screened for additional quality control specific to this analysis. Notably, data that suggested tower wake effects from the booms were removed by identifying the boom orientation and removing any associated data within 60 degrees (+/- 30 degrees) of the boom orientation. These screening processes collectively removed 3.2% of the 20 Hz data. Block-averaged fluxes (Babić et al., 2016; Wildmann et al., 2019) were then calculated using an eddy-covariance approach (Aubinet et al., 2012) from the surviving data for all heights using two different Reynolds decomposition times, discussed below in "Reynolds Decomposition Time". The stability and turbulence metrics were calculated from these fluxes.

### 2.2.1 Metric selection

This analysis evaluates three stability metrics – the Obukhov length ($L$), turbulence kinetic energy ($TKE$), and the turbulence dissipation rate ($\epsilon$). While many other stability metrics exist (Stull, 1988), this metric subset was selected to include a mix of both categorical and qualitative metrics and only metrics with a strong and consistent diurnal cycle.

The mix of categorical and qualitative metrics provides a means to to categorize stability behavior based on regime. For example, because the Obukhov length categorically represents atmospheric stability (i.e., stable vs. unstable), we can compare the values of another stability metric ($TKE$, for example) during periods where the Obukhov length is stable with those during periods where the Obukhov length is unstable.

The diurnal cycle restriction was designed to assess how well the metric captures diurnal variability in boundary layer behavior. For $L$, each period was classified with a stability based on Table 2. Then, for each hour of the day, the total number of cases for each stability bin was summed and then normalized by the total number of cases for that hour throughout the experiment. The $TKE$ and $\epsilon$ diurnal cycles were analyzed as diurnal distributions. For all metrics, the diurnal test was applied to the data from the surface (10 m) and aloft (100 m) separately and evaluated qualitatively. Turbulence intensity, which is another common metric in the wind energy community, was shown to have a variable and inconsistent diurnal cycle and was excluded through this screening.

### 2.2.2 Obukhov length ($L$)

The Obukhov length, $L$, is proportional to the height above the surface at which buoyant factors first dominate over mechanical shear production of turbulence:

$$L = \frac{-\overline{\theta_v} u_*^3}{\kappa g \overline{w'\theta_v'}} \tag{1}$$





where $\kappa = 0.4$ is the von Kármán constant; $g = 9.81$ m s$^{-2}$ is the acceleration due to gravity; $\overline{w'\theta'_v}$ represents the heat flux (K m s$^{-1}$); $\overline{\theta_v}$ represents the average virtual temperature (K); and $u_*$ denotes the friction velocity (m s$^{-1}$):

$$u_* = (\overline{u'w'}^2 + \overline{v'w'}^2)^{1/4} \tag{2}$$

(Monin and Obukhov, 1954). Both the heat flux and the friction velocity were calculated based on the eddy-covariance approach using the appropriate averaging window, and the heat flux was calculated using the sonic temperature, as in Burns et al. (2012).

$\theta_v$ is defined as:

$$\theta_v = T_v (\frac{p_0}{p})^{R/C_p} \tag{3}$$

where $T_v$ is the virtual temperature (K); $p_0$ is a reference pressure (Pa); $p$ is the pressure at a given height (Pa); and $R/C_p \approx 0.286$. Herein, $\theta_v$ was approximated as the air temperature due to a lack of localized pressure measurements. Further, because not all 47 towers had temperature measurements available at 10 m, the virtual potential temperature for the horizontal

homogeneity analysis (described later) was assumed to be a uniform 300 K. The $L$ values were binned according to a schema adapted from Gryning et al. (2007) (Table 2).

**Table 2.** Stability parameter binning classifications adapted from Gryning et al. (2007)

| Category | Range |
| --- | --- |
| Very Stable | $0 \leq L < 200$ |
| Stable | $200 \leq L < 500$ |
| Near Neutral | $|L| \geq 500$ |
| Unstable | $-500 < L \leq -200$ |
| Very Unstable | $-200 < L \leq -0$ |

### 2.2.3 Turbulence kinetic energy ($TKE$)

$TKE$ quantifies turbulence in the flow via covariances. The mean $TKE$, $\overline{TKE}$ (m$^2$ s$^{-2}$), is described by:

$$\overline{TKE} = \frac{1}{2}[\overline{u'^2} + \overline{v'^2} + \overline{w'^2}] \tag{4}$$

where $u'$, $v'$, and $w'$ (m s$^{-1}$) are the Reynolds perturbation components from the mean wind (Stull, 1988). In all cases, perturbations are based on a block averaging window over the relevant time period.

### 2.2.4 Turbulence dissipation rate ($\epsilon$)

The turbulence dissipation rate, $\epsilon$ (m$^2$ s$^{-3}$), represents the rate at which $TKE$ is dissipated or converted into heat (Stull, 1988) and was calculated using the structure function method (Piper and Lundquist, 2004; Muñoz-Esparza et al., 2018; Wildmann





et al., 2019). According to this method, $\epsilon$ is related to a second-order structure function, $D_U(\tau)$ by the following relation:

$$\epsilon = \frac{1}{U\tau}[aD_U(\tau)]^{\frac{3}{2}} \tag{5}$$

where $U$ is the horizontal velocity (m s$^{-1}$), $a = 0.52$ is the Kolmogorov constant, $\tau$ is the temporal separation (s), and $D_U(\tau)$ is the structure function:

$$D_U(\tau) = <[U(t+\tau) - U(t)]^2> \tag{6}$$

where $<>$ denotes the ensemble average and $\tau$ was treated as 2 s, consistent with prior analyses (Bodini et al., 2018; Wildmann et al., 2019; Bodini et al., 2019). To calculate $\epsilon$, every 30 s, a centered 2 min window of data was targeted. The structure function was applied to this 2 min window, and the average value of the relevant structure function was retained for each $\epsilon$ calculation. These data were maintained at a 30 s resolution both because $\epsilon$ varies on a shorter timescale than $L$ or $TKE$ and is also not defined by a Reynolds decomposition window.

### 2.2.5 Reynolds decomposition time

Both $L$ and $TKE$ require Reynolds decomposition calculations (Stull, 1988; Reynolds, 1895). In a Reynolds decomposition, any variable $\psi$ is split into a mean $\overline{\psi}$ and a turbulent component $\psi'$. The separation between these two components is determined by the Reynolds decomposition averaging window. Because determining a Reynolds decomposition window that appropriately differentiates mean behavior from turbulent motions is consequential to the resultant stability characterization, many methods
to determine the Reynolds decomposition window exist.

Both $L$ and $TKE$ decomposition windows were calculated according to both the ogive method (Babić et al., 2012) (which resulted in a 30 min window) and a 10 min window based on wind energy industry conventions (IEC). While we also considered using a variable window based on a multi-flux-resolution decomposition (MRD) approach (Howell and Mahrt, 1997), a previous investigation of the Perdigão dataset with the MRD determined that the MRD was insufficient, so we instead adopted
a consistent 30 min averaging approach for this dataset (Mosso et al., 2025).

An appropriate averaging window was first determined using an ogive method as described in Desjardins et al. (1989), Oncley et al. (1996), and Babić et al. (2012). This assessment was performed to determine whether a uniform averaging period could be utilized for heat fluxes $(\overline{w'\theta'_v})$ and friction velocities $(\overline{u_*})$ across various heights, times of day, and tower locations. The ogive method requires calculation of fluxes using several averaging periods and then defines an appropriate averaging
period based on the frequency in which there are no more (or negligible) contributions to a flux covariance. By evaluating the cumulative eddy contribution according to increasing averaging periods, an asymptote separates the local turbulent fluctuations from the larger-scale (mesoscale) fluctuations. For this dataset, heat flux and friction velocity values were calculated at each height for each tower location according to 1, 5, 10, 20, 30, and 60 min Reynolds decomposition windows. All values for a given Reynolds decomposition period were then averaged to a single value for that averaging period. To differentiate between
times of day, the heat flux values were subset to 00–04 UTC and 12–16 UTC to represent nighttime (assumed stable) and daytime (assumed unstable) values, respectively. Based on this analysis, we used a consistent 30 min Reynolds decomposition window for both stable and unstable cases at all towers.





Secondly, we considered the wind energy industry standard, which uses a constant 10 min averaging window (Bailey et al., 1997; IEC). This standard assumes a clear separation or "spectral gap" (Hoven, 1957) between the large-scale mesoscale

motions and the small-scale turbulence, which does not always exist (Larsén et al., 2013; Kang and Won, 2016; Larsen et al., 2018). The 10 min averaging is intended to target only turbulence (Larsén et al., 2013; Kang and Won, 2016; Larsen et al., 2018).

## 2.3 Postprocessing

Extrema were also screened out of the stability metric calculations. This screening process involved isolating the distribution of

values for each stability metric according to a given Reynolds decomposition window and replacing values outside of a given threshold as NaN (not a number). A common 99.5th percentile threshold was established to simultaneously remove anomalous data and protect representative data. Further, time periods flagged according to one stability metric were not necessarily flagged according to another, ensuring that the representativeness of each stability metric could be evaluated independently.

The process to apply this extrema threshold was tailored to each metric. $L$ extrema were flagged by extrema in $u_*$, $\overline{w't'_c}$ and

$\overline{\theta_v}$ (Eq. 1). This decision to flag the constituent variables of the Obukhov length as opposed to the Obukhov length itself both ensured consistency across all stages of the calculations and also avoided targeted removal of near-neutral cases. Collectively, this Obukhov length screening process removed 1.06% of $L$ values.

$u_*$ extrema were identified as $u_*$ values greater than the 99.5% of the 30 min $u_*$ distribution (1.20 m s$^{-1}$). The same (i.e., 1.20 m s$^{-1}$) 30 min $u_*$ threshold was also applied to the 10 min $u_*$ distribution to avoid mischaracterizing larger 10 min $u_*$

values as extrema.

$\overline{w't'_c}$ extrema were identified according to a process similar to that used to identify extrema in $u_*$, although the $\overline{w't'_c}$ extrema were flagged on both sides of the distribution. The 30 min distribution determined both the 30 min and 10 min threshold for $\overline{w't'_c}$, as was the case with $u_*$. However, in contrast, $\overline{w't'_c}$ extrema were identified by flagging the most extreme 0.5th percentile of the positive and negative $\overline{w't'_c}$ values separately. This distinction based on the heat flux sign ensured that both stable and

unstable cases were screened and that these screenings were independent. This screening identified data greater than 0.521 K m s$^{-1}$ and less than -0.226 K m s$^{-1}$. These screening values correspond to 633.6 W m$^{-2}$ and -274.4 W m$^{-2}$, assuming that the product of the density of moist air, $\rho_{air}$, and the specific heat of air, $C_p$, is 1216 W m$^{-2}$ (K m s$^{-1}$)$^{-1}$ (Stull, 1988).

$\overline{\theta_v}$ extrema were also flagged on both sides of the distribution, although the underlying data considered for the $\overline{\theta_v}$ process differed from those considered for the $u_*$ and $\overline{w't'_c}$ processes. While the $u_*$ and $\overline{w't'_c}$ screening processes considered data from

all 47 towers, the $\overline{\theta_v}$ screening process only considered temperature measurements from the three 100 m towers to determine a reliable screening threshold. This distinction was made to account for the inconsistent temperature availability across the site. The net result of this $\overline{\theta_v}$ screening was that $\overline{\theta_v}$ less than 280.47 K or greater than 306.5 K were flagged.

The $TKE$ extrema screening process was most analogous to that employed for $u_*$. $TKE$ greater than the 99.5th percentile of the 30 min distribution (5.83 m$^2$ s$^{-2}$) were flagged. Because the $TKE$ screening depended only on $TKE$ itself, 0.37%

of $TKE$ values were flagged. Finally, the $\log_{10}\epsilon$ extrema screening process removed data from the top (-0.509) and bottom (-5.45) 0.5th percentiles, again constituting 1% of $\log_{10}\epsilon$ values.





## 2.4 Stability assessment

The variability in atmospheric stability was evaluated with two criteria: hub-height predictive index (HHPI) and horizontal homogeneity (HoH). These criteria, described more fully below, collectively describe the sensitivity of stability in time, altitude, and spatial location.

Atmospheric stability was first evaluated based on the HHPI. The HHPI assessment was designed to assess how well surface-level classifications predicted classifications at a representative wind turbine hub height of 100 m. This assessment could provide insight into whether surface measurements could predict hub height behavior without the investment of a full 100 m tower. The $L$ HHPI was calculated as the fraction of cases in which the surface value and the aloft value showed the same stability classification. The $TKE$ and $\epsilon$ HHPI were represented by the linear regression $R^2$ between the surface and aloft values for each metric. The HHPI assessment was also extended to other intermediate heights present on the towers (i.e., 20, 40, 60, 80, and 100 m). For this extension of the HHPI analysis, the regression or agreement classification was performed by treating each of the intermediate heights as the base height compared to the 100 m hub-height calculations.

Each metric was also evaluated for HoH. This assessment was designed to quantify the smallest number and location of meteorological towers necessary to capture the site's variability in surface (10 m) stability. For each metric, all 47 towers with sonic anemometers were considered. Because only 18 of these towers had available 10 m temperature measurements, the virtual potential temperature was assumed to be a uniform 300 K for the Obukhov length calculation just for the HoH analysis. Thus, this analysis could not account for differences in virtual potential temperature that might occur between towers. These 47 towers were then represented as a graph network, with a node for each tower location and the edge weights as the surface stability correlations. The smallest number of meteorological towers were then determined based on the Louvain community detection algorithm (Blondel et al., 2008), as implemented by the NetworkX Python package (Hagberg et al., 2008). The Louvain algorithm determined which towers were redundant by identifying tower subgroups and assigning each tower to one of these subgroups. For this analysis, the number of subgroups corresponded to the number of necessary towers and towers within a given subgroup were considered redundant. Further, the Louvain community detection algorithm was chosen because of its (non)treatment of overlapping communities, further ensuring that each community is treated independently with a unique tower.

## 3 Results

### 3.1 Site characterization

#### 3.1.1 Ogive analysis

We recommend a uniform 30 min Reynolds decomposition window based on the results from the ogive analysis. Based on the ogives, a 30 min heat flux averaging period is supported for all heights and tower locations (Fig. 2a, d, and g). The heat flux ogive at all three tower locations show an asymptote between an averaging period of 20–30 min and a shift to mesoscale





fluctuations at an averaging period of 60 min. The heat flux ogives also reflect ridge/valley distinctions. The heat flux ogives for the two ridges (Fig. 2a and g) show both higher values and greater spread than the heat flux ogive for the valley (Fig. 2d).



**Figure 2.** Heat flux ogives at each 100 m tower (tse04 (SW ridge): a, b, c; tse09 (valley): d, e, f; tse13 (NE ridge): g, h, i) and each available height, for all (a, d, g), stable (0–4 UTC, b, e, h) and unstable (12–16 UTC, c, f, i) cases. Points represent the mean value for a tower/height combination and error bars represent the standard error. (a) tse04 overall (SW ridge), (b) tse04 (SW ridge) stable, (c) tse04 (SW ridge) unstable, (d) tse09 (valley) overall, (e) tse09 (valley) stable, (f) tse09 (valley) unstable, (g) tse13 (NE ridge) overall, (h) tse13 (NE ridge) stable, (i) tse13 (NE ridge) unstable.

275    A 30 min heat flux averaging period is also appropriate for stable (00–04 UTC) cases at all three 100 m towers (Fig. 2b, e, and h). The stable heat flux ogives (Fig. 2b, e, and h) show an asymptote at small averaging periods at all tower locations. The





stable ogive on the NE ridge (Fig. 2h) shifts to mesoscale fluctuations at 60 min, and the stable ogive on the SW ridge (Fig. 2b) shows a less-pronounced mesoscale shift at 60 min. The stable ogives also reflect ridge/valley distinctions. The stable ogives on the two ridges (Fig. 2b and h) show larger values and a larger spread than the stable ogive for the valley (Fig. 2e), but the 30 min period is still appropriate.

A 30 min heat flux averaging period is also justified for unstable (12–16 UTC) cases (Fig. 2c, f, and i). The unstable heat flux ogives at all three tower locations show asymptotic behavior that includes 30 min. The unstable ogives also reflect ridge/valley distinctions. The unstable ogives on the two ridges (Fig. 2c and i) show larger values and a larger spread than the unstable ogive for the valley (Fig. 2f).

A 30 min friction velocity averaging period is generally justified across heights for the NE ridge and valley (Fig. 3d and g). All friction velocity ogives for the valley (Fig. 3d) and NE ridge (Fig. 3g) and all but one ogive for the SW ridge (Fig. 3a) show an asymptote at a 30 min averaging period. The friction velocity ogive at 10 m on the SW ridge (Fig. 3a) does not asymptote at 30 min. The friction velocity ogives (Fig. 3a, d, and g) also differ by location. The friction velocity ogive in the valley (Fig. 3d) has the largest spread between heights of the three locations.

A 30 min friction velocity averaging period is also generally justified for stable (00–04 UTC) cases for the valley (Fig. 3e) and for the NE ridge (Fig. 3h). The stable friction velocity ogives for these two locations show an asymptote for a 30 min averaging period and shift toward mesoscale fluctuations for a 60 min averaging period. One location that may suggest a need for a longer averaging period during stable cases is the SW ridge (Fig. 3b). The stable friction velocity ogives on the SW ridge (Fig. 3b) do not asymptote at 30 min, and the ogives continue to increase beyond a 60 min averaging period.

A 30 min friction velocity averaging period is also supported for all tower locations and heights during unstable (12–16 UTC) cases (Fig. 3c, f, and i). The unstable friction velocity ogives at all tower locations show asymptotic behavior for a 30 min averaging period and a shift to mesoscale fluctuations for a 60 min averaging period. The unstable friction velocity ogives also reflect differences based on tower location. The unstable friction velocity ogive for the SW ridge (Fig. 3c) shows similar behavior across heights while the unstable friction velocity ogives for the valley (Fig. 3f) and NE ridge (Fig. 3i) show greater spread between heights.

### 3.1.2 Wind roses and their spatial and vertical variability

The 30 min wind roses for the three 100 m towers differ between the valley (Fig. 4c and d) and the ridges (Fig. 4a, b, e, and f). Both the SW (Fig. 4a) and NE (Fig. 4e) ridge surface winds show primarily a southwesterly flow. In contrast, the surface valley winds (Fig. 4c) show primarily a southeasterly flow. Further, the surface valley winds (Fig. 4c) are slower than the surface winds on either of the two ridges (Fig. 4a and e), likely due to sheltering effects.

The wind roses for the three 100 m towers also differ between the 10 m surface winds and the 100 m winds. The 100 m winds at all three tower locations (Fig. 4b, d, and f) are more diffuse than the winds at their respective 10 m locations. The 100 m winds at all three tower locations are also generally faster and more consistent than the winds at their respective 10 m locations. These site and height distinctions between wind roses are also preserved, regardless of the averaging window (not shown).



**Figure 3.** $u_*$ ogives at each 100 m tower (tse04 (SW ridge): a, b, c; tse09 (valley): d, e, f; tse13 (NE ridge): g, h, i) and each available height, for all (a, d, g), stable (0–4 UTC, b, e, h) and unstable (12–16 UTC, c, f, i) cases, as in Fig. 2. Points represent the mean value for a tower/height combination and error bars are described with the standard error.

### 3.1.3 Diurnal variability

All three metrics exhibit a strong diurnal cycle (Figs. 5–12) as would be expected for this onshore mid-latitude region in late spring/early summer.

The $L$ diurnal cycle (Fig. 5) differs between the ridges (Fig. 5a, b, e, f) and the valley (Fig. 5c, d). The $L$ diurnal cycle shows

mostly Stable (S) and Very Stable (VS) cases in the early morning (00–05 UTC) and evening (21–00 UTC) hours, with a shift toward mostly Unstable (U) and Very Unstable (VU) cases in the middle of the day (06–20 UTC) at all locations (Fig. 5). The





valley $L$ diurnal cycle is unique in its larger relative percentage of near-neutral cases aloft (Fig. 5d). While the $L$ diurnal cycle on the NE ridge (Fig. 5f) does contribute near-neutral cases aloft, this contribution is less extreme than that seen in the valley. Further, the $L$ diurnal cycle on the SW ridge has negligible near-neutral cases aloft (Fig. 5b). The $L$ diurnal cycle in the valley 320 is also unique in its reduction of the relative contribution of VU cases from the surface (Fig. 5c) to aloft (Fig. 5d). Again, while the $L$ diurnal cycle on the NE ridge has more VU cases at the surface (Fig. 5e) than aloft (Fig. 5f) throughout all daytime hours, the $L$ diurnal cycle on the SW ridge shows more similar frequencies of VU cases between the surface (Fig. 5a) and aloft (Fig. 5b) in the morning, with fewer VU cases at 100 m in the later afternoon. Of note, the SW ridge has fewer VU cases than either the valley or the NE ridge.

The $u_*$ diurnal cycle (Fig. 6) explains the site-based differences in the reduction in VU contribution. $u_*$ shows smaller values in the early morning (00–05 UTC) and evening (19–00 UTC) hours, with a shift toward larger values in the middle of the day (06–20 UTC) at all heights. The middle-of-day $u_*$ in the valley also increases from 10 m measurements (Fig. 6c) to measurements at 100 m (Fig. 6d), suggesting an increased role of mechanical forcing and therefore a decreased role of buoyant forcing. This shift corresponds to a reduction in the representation of VU cases from the surface (Fig. 6c) to aloft (Fig. 6d). A 330 similar trend exists at the NE ridge, although not as strong. Again, the middle-of-day $u_*$ increase from the surface (Fig. 6e) to aloft (Fig. 6f) corresponds to a reduced contribution from VU cases. In contrast, the middle-of-day $u_*$ at the SW ridge does not increase from the surface (Fig. 6a) to aloft (Fig. 6b) but actually slightly decreases. This middle-of-day $u_*$ decrease from the surface to aloft along the SW ridge suggests a reduced role of mechanical forcing and therefore an increased role of buoyant forcing, thereby supporting the consistent number of VU cases between surface and aloft at the SW ridge in the morning hours 335 (9–12 UTC).

In contrast, the heat flux $(\overline{w't'_c})$ diurnal cycle (Fig. 7) does not explain the reduced VU contribution aloft in the valley. The heat flux is smaller in the early morning (00–05 UTC) and evening (19–00 UTC) hours, with a shift and grows in the middle of the day (06–20 UTC). The midday maximum in heat flux along the two ridges is larger at the surface (Fig. 7a and e) than aloft (Fig. 7b and f). Further, in the valley, the midday heat flux maximum grows from the surface (Fig. 7c) to aloft (Fig. 7d).

Further, the $u_*$ diurnal cycle appears regardless of the decomposition period. $u_*$ is smallest during the morning hours (00–05 UTC), peaks in the middle of the day (06–20 UTC), and then drops again in the evening (21–24 UTC), regardless of the Reynolds decomposition period. During unstably stratified periods, $u_*$ according to the 10 min Reynolds decomposition period is smaller than $u_*$ based on a 30 min Reynolds decomposition period. This inability of the 10 min Reynolds decomposition period to resolve the full flux is most apparent at 10 m on the SW ridge (Fig. 8a).

The maximum $u_*$ found in the stable periods of the $u_*$ diurnal cycle also vary between the Reynolds decomposition periods. During these stable periods, the 30 min averaging approach yields the highest $u_*$, followed by the 10 min averaging approach. Again, this discrepancy at the surface is largest on the SW ridge (Fig. 8a). In contrast, the valley aloft (Fig. 8d) shows larger $u_*$ differences based on the Reynolds decomposition period during stable periods than the two ridges (Fig. 8b, f).

The $TKE$ diurnal cycle also shows smaller values in the early morning (00–05 UTC) and evening (21–00 UTC) hours and 350 grows in the middle of the day (06–20 UTC) (Fig. 9). While all three tower locations show larger values aloft (Fig. 9b, d, and f) than at the surface (Fig. 9a, c, and e), this trend is especially pronounced in the valley (Fig. 9b and c), perhaps due to advection





of surface-generated turbulence from the ridges. Further, the valley shows a more muted diurnal cycle at 10 m (Fig. 9c) than the ridges show at 10 m (Fig. 9a and e).

The $TKE$ diurnal cycle also emerges regardless of the decomposition approach, although the maximum value of $TKE$ is
very sensitive to the averaging period used. $TKE$ is smaller during the early morning and evening hours and grows in the middle of the day for all Reynolds decomposition windows, as was the case with $u_*$ (Fig. 10) and as expected over land. Also consistent with the behavior observed with the $u_*$ diurnal cycle, during unstably stratified periods, 30 min $TKE$ and 10 min $TKE$ are smaller (Fig. 10). Further, this unstable period discrepancy based on the Reynolds decomposition window is again most pronounced at the surface on the SW ridge (Fig. 10a) and aloft in the valley (Fig. 10d). At the same time, the $TKE$
diurnal cycle (Fig. 10) varies more during stable periods based on the Reynolds decomposition window than the $u_*$ diurnal cycle (Fig. 8). This spread based on the Reynolds decomposition window observed in the $TKE$ emerges at both altitudes for all three towers (Fig. 10).

These trends in the $TKE$ diurnal cycle also occur at intermediate heights (Fig. 11) with the largest differences at 10 m in the valley and on the canopied NE ridge. The $TKE$ in the valley (Fig. 11c, d) shows stratified layers during nighttime
hours, with smaller values at the surface and larger values with increasing height, suggesting decoupling or top-down boundary layer forcing. During the daytime period, the valley site shows stratification as well. This $TKE$ stratification in the valley also persists for both Reynolds decomposition approaches. The 30 min valley $TKE$ (Fig. 11c) differs most strongly between 10 m and 100 m, especially during the unstable middle of the day (Fig. 11d). The 10 min valley $TKE$ (Fig. 11e) reduces the peak $TKE$ and also reduces the gap between 10 m and 100 m $TKE$. The $TKE$ along the two ridges (Fig. 11a, b, e, f) does
not show stratified layers. Rather, multiple levels frequently show the same value of $TKE$, suggesting well-mixed boundary layers. Further, the $TKE$ along the SW ridge, with less vegetation (Fig. 11a, b), is larger at the surface than aloft. In contrast, during the middle of the day, the NE ridge (Fig. 11e, f) with extensive vegetation shows smaller $TKE$ at 10 m than at upper levels.

The $\epsilon$ diurnal cycle is the least pronounced of the three metrics considered (Fig. 12), which has implications for numerical
modeling of flow in this complex terrain. While generally smaller values occur in the morning (00–05 UTC) and evening (21–00 UTC) hours and larger values in the middle of the day (06–20 UTC), the diurnal cycle is muted both at the surface on the NE ridge (Fig. 12e) and aloft in the valley (Fig. 12d). The $\epsilon$ diurnal cycle also differs between the ridges (Fig. 12a, b, e, and f) and the valley (Fig. 12c and d). While the two ridges show smaller $\epsilon$ values aloft (Fig. 12b and f) than at the surface (Fig. 12a and e) (opposite the $TKE$ trends), the valley shows relatively consistent $\epsilon$ values between the surface (Fig. 12c) and aloft (Fig. 12d).
The valley's larger values may again be due to advection of turbulence from the neighboring ridges, which must then dissipate in the valley. This local imbalance of dissipation rate, as seen also in Wildmann et al. (2019), has implications for numerical models that assume local balance between production and dissipation.





### 3.2 Stability metric analysis

#### 3.2.1 Hub-height predictive index

Perdigão surface (10 m) measurements fail to represent conditions at turbine hub height (100 m). All metrics show poor levels of HHPI at the surface. The surface $L$ rate of agreement is 50% or lower for all three towers (Fig. 13). The surface $L$ HHPI also varies between towers, with higher HHPI on the two ridges than in the valley.

For $TKE$, surface HHPI is quantified by correlation and shows a poor $R^2$ for all tower locations with variability between towers. The $TKE$ surface HHPI is highest on the NE ridge (0.618) (Fig. 13) and in the valley (0.616) (Fig. 13). The SW ridge
has a lower $TKE$ surface HHPI of 0.509 (Fig. 13).

The $\epsilon$ surface HHPI also shows very poor performance (Fig. 13) with $R^2$ values always less than 0.4. This consistently low surface HHPI leaves $\epsilon$ as the worst metric to make hub-height predictions based on surface measurements. $\epsilon$ surface HHPI also reflects ridge/valley distinctions with higher surface HHPI along the two ridges (Fig. 13a, c) than in the valley (Fig. 13b).

Higher locations on the towers lead to higher HHPI (Fig. 13). The $TKE$ HHPI profile on the SW ridge (Fig. 13a) improves
agreement with the 100 m measurement as measurements move up. The $TKE$ HHPI profiles in the valley (Fig. 13b) and on the NE ridge (Fig. 13c) consistently improve hub-height agreement with higher observations. The $\epsilon$ HHPI profile shows the most consistent increase across heights as well as the greatest consistency across tower locations. Although the $\epsilon$ HHPI (Fig. 13) at the surface is the lowest of the three metrics, the $\epsilon$ HHPI (Fig. 13) steadily increases to overtake the $L$ HHPI (Fig. 13) at all three sites.

HHPI is negligibly affected by the Reynolds decomposition period. While assessments like the $L$ HHPI on the SW ridge (Fig. 14a) and the $TKE$ HHPI both on the SW ridge (Fig. 14d) and in the valley (Fig. 14e) suggest slight differences between Reynolds decomposition periods, these differences are both subtle and unsustained at all tower measurement heights. This lack of performance difference between the 10 min and 30 min Reynolds decomposition windows does not necessarily suggest that 10 min Reynolds decomposition windows and 30 min Reynolds decomposition windows resolve similar fluxes. In fact, several
analyses in this work, including ogives (Fig. 2) and diurnal cycles (Fig. 5–Fig. 12), suggest otherwise. Rather, because the HHPI assessment compares stability metrics with the same Reynolds decomposition window, the lack of performance improvement with a 30 min Reynolds decomposition window may instead suggest that a longer Reynolds decomposition window may not be an appropriate alternative to building taller towers.

#### 3.2.2 Horizontal homogeneity

We sought to define the minimum number of towers to represent the variability in the 10 m measurements at Perdigão using the Louvain community detection algorithm. A nine-tower subset spans all groupings across all metrics (Fig. 15). While this subset is not the only potential subset that spans all groupings across all metrics, this tower subset indicates the smallest number of towers necessary to fully characterize the site's partitions in $TKE$, $L$, and $\epsilon$.

The towers cluster into groups that generally follow terrain boundaries (Fig. 15). For example, with 30 min averaging, the $L$
groupings (Fig. 15a) define the SW (purple) and NE (orange) ridges, respectively, with the exception of the forest green rne02





(Fig. 15a). The ridge partitions are also influenced by two transect (brown and light green) partitions (Fig. 15a). In between the ridge partitions and the transect groupings lies a valley partition (Fig. 15a, dark blue) and a valley pocket (Fig. 15a, lavender). Similar terrain patterns are reflected in the 10 min $L$ groupings (Fig. 15b), the $TKE$ (Fig. 15c, d) groupings, and the $\epsilon$ (Fig. 15e, f) groupings.

$TKE$ groupings and $\epsilon$ groupings also differ from the $L$ groupings. While the $L$ groupings treat rne02 (Fig. 15a, forest green) as its own partition, the $TKE$ groupings (Fig. 15c) see this tower as part of the NE ridge (orange), and the $\epsilon$ groupings (Fig. 15e) instead treat the tower as an extension of the SW ridge (purple). The $TKE$ groupings (Fig. 15c) also suggest a new (turquoise) partition outside of the SW ridge that neither the $L$ (Fig. 15a) nor the $\epsilon$ groupings (Fig. 15e) acknowledge. Finally, the $TKE$ groupings and the $\epsilon$ groupings differ in their characterization of tse08. The $TKE$ groupings treat tse08 (Fig. 15c,

middle purple) as an extension of the SW ridge while the $\epsilon$ groupings absorb tse08 into the valley (Fig. 15e, dark blue). The $L$ characterization of this tower is different still, treating this tower as an extension of the outside (brown) transect (Fig. 15a).

The Reynolds decomposition window shifts the placement – but not the number – of partitions for a given metric. Seven $L$ groupings exist, regardless of whether a 30 min (Fig. 15a) or 10 min (Fig. 15b) Reynolds decomposition window is employed. However, the L groupings with a 30 min Reynolds decomposition window (Fig. 15a) treat rne02 (dark green) as its own

partition while the $L$ groupings with a 10 min Reynolds decomposition window (Fig. 15b) treat rne02 as part of the NE ridge. The $L$ groupings with the 10 min Reynolds decomposition window (Fig. 15b) also treat tse08 (light blue) as its own partition while the $L$ groupings with the 30 min Reynolds decomposition window treat tse08 as part of the (brown) transect grouping. The $L$ grouping with the 30 min Reynolds decomposition window (Fig. 15a) also treats v01 (lavender) as its own partition while the $L$ groupings with the 10 min Reynolds decomposition window treat v01 as an extension of the (light green) transect

grouping. The net result of these shifts in partition placement is that the same number of towers is necessary to characterize $L$ for a given Reynolds decomposition window, but two additional towers are necessary to capture the variability between metrics. The Reynolds decomposition window also shifts the placement of the five $TKE$ partitions. Tower tse08 (Fig. 15c, middle purple) is treated as an extension of the SW ridge with a 30 min Reynolds decomposition window and part of the valley with a 10 min Reynolds decomposition window.

**Figure 4.** Perdigão field campaign wind rose with 30 min averaging at the three 100 m towers and at 10 m (a, c, e) and 100 m (b, d, f) altitudes. The rings (i.e., 5, 15, 25, 35) correspond to the percentage of data that aligns with a given speed and direction. (a) tse04 (SW ridge) at 10 m, (b) tse04 (SW ridge) at 100 m, (c) tse09 (valley) at 10 m, (d) tse09 (valley) at 100 m, (e) tse13 (NE ridge) at 10 m, (f) tse13 (NE ridge) at 100 m.



**Figure 5.** *L* diurnal cycle for full campaign period with 30 min averaging: (a) tse04 (SW ridge) at 10 m, (b) tse04 (SW ridge) at 100 m, (c) tse09 (valley) at 10 m, (d) tse09 (valley) at 100 m, (e) tse13 (NE ridge) at 10 m, (f) tse13 (NE ridge) at 100 m.





**Figure 6.** $u_*$ diurnal cycle for full campaign period. Red lines indicate the median, and the box and whiskers are defined based on Q1 (25th percentile), Q3 (75th percentile), and the interquartile range (IQR) (Q3–Q1). The box encloses the IQR and the whiskers extend to $Q1 - 1.5 * IQR$ and $Q3 + 1.5 * IQR$. (a) tse04 (SW ridge) at 10 m, (b) tse04 (SW ridge) at 100 m, (c) tse09 (valley) at 10 m, (d) tse09 (valley) at 100 m, (e) tse13 (NE ridge) at 10 m, (f) tse13 (NE ridge) at 100 m.





**Figure 7.** Heat flux diurnal cycle for full campaign period with 30 min averaging. Red lines indicate the median, and the box and whiskers are defined based on Q1 (25th percentile), Q3 (75th percentile), and the interquartile range (IQR) (Q3–Q1). The box encloses the IQR and the whiskers extend to $Q1 - 1.5 * IQR$ and $Q3 + 1.5 * IQR$. (a) tse04 (SW ridge) at 10 m, (b) tse04 (SW ridge) at 100 m, (c) tse09 (valley) at 10 m, (d) tse09 (valley) at 100 m, (e) tse13 (NE ridge) at 10 m, (f) tse13 (NE ridge) at 100 m.



**Figure 8.** $u_*$ diurnal cycle for full campaign period according to three decomposition intervals. The solid line indicates the median value, and the band indicates the standard error. (a) tse04 (SW ridge) at 10 m, (b) tse04 (SW ridge) at 100 m, (c) tse09 (valley) at 10 m, (d) tse09 (valley) at 100 m, (e) tse13 (NE ridge) at 10 m, (f) tse13 (NE ridge) at 100 m.

**Figure 9.** $TKE$ diurnal cycle for full campaign period with 30 min averaging. Red lines indicate the median, and the box and whiskers are defined based on Q1 (25th percentile), Q3 (75th percentile), and the interquartile range (IQR) (Q3–Q1). The box encloses the IQR and the whiskers extend to $Q1 - 1.5 * IQR$ and $Q3 + 1.5 * IQR$. (a) tse04 (SW ridge) at 10 m, (b) tse04 (SW ridge) at 100 m, (c) tse09 (valley) at 10 m, (d) tse09 (valley) at 100 m, (e) tse13 (NE ridge) at 10 m, (f) tse13 (NE ridge) at 100 m.







**Figure 10.** $TKE$ diurnal cycle at three towers (tse04: a, b; tse09: c, d; tse13: e, f) for the full campaign period according to two averaging approaches at two heights (10 m: a, c, e; 100 m: b, d, f). The solid lines indicate the median value, and the bands indicate the standard error. (a) tse04 (SW ridge) at 10 m, (b) tse04 (SW ridge) at 100 m, (c) tse09 (valley) at 10 m, (d) tse09 (valley) at 100 m, (e) tse13 (NE ridge) at 10 m, (f) tse13 (NE ridge) at 100 m.





**Figure 11.** Diurnal cycle of $TKE$ at all heights for each 100 m tower (tse04: a and b; tse09: c and d; tse13: g and h) according to two averaging approaches (30 min: a, c, e; 10 min: b, d, f). The solid line indicates the median value, and the band is the standard error. (a) tse04 (SW ridge) 30 min, (b) tse04 (SW ridge) 10 min, (c) tse09 (valley) 30 min, (d) tse09 (valley) 10 min, (e) tse13 (NE ridge) 30 min, (f) tse13 (NE ridge) 10 min.



**Figure 12.** $\epsilon$ diurnal cycle for full campaign period. Red lines indicate the median, and the box and whiskers are defined based on Q1 (25th percentile), Q3 (75th percentile), and the IQR (Q3–Q1). The box encloses the IQR and the whiskers extend to $Q1 - 1.5 * IQR$ and $Q3 + 1.5 * IQR$. (a) tse04 (SW ridge) at 10 m, (b) tse04 (SW ridge) at 100 m, (c) tse09 (valley) at 10 m, (d) tse09 (valley) at 100 m, (e) tse13 (NE ridge) at 10 m, (f) tse13 (NE ridge) at 100 m.

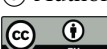

WIND
ENERGY
SCIENCE
DISCUSSIONS

**Figure 13.** Correlation coefficient between the base height and 100 m for full campaign period with 30 min averaging. (a) tse04 (SW ridge), (b) tse09 (valley), (c) tse13 (NE ridge). tse13 60 m $L$ calculations were omitted due to high (76%) temperature data unavailability.



**Figure 14.** Correlation coefficient between the base height and 100 m for full campaign period according to two averaging approaches. In all cases, $\epsilon$ calculations are maintained at their 30 s resolution. (a) tse04 (SW ridge) $L$, (b) tse04 (SW ridge) $TKE$, (c) tse04 (SW ridge) $\epsilon$, (d) tse09 (valley) $L$, (e) tse09 (valley) $TKE$, (f) tse09 (valley) $\epsilon$, (g) tse13 (NE ridge) $L$, (h) tse13 (NE ridge) $TKE$, (i) tse13 (NE ridge) $\epsilon$.







**Figure 15.** Groupings determined by the Louvain community detection algorithm for each metric. The number of colors corresponds to the number of tower groupings, and towers of the same color correspond to the same tower grouping. A nine-tower subset indicated with stars spans all groupings across all metrics. (a) $L$ 30 min, (b) $L$ 10 min, (c) $TKE$ 30 min, (d) $TKE$ 10 min, (e) $\epsilon$ 30 s.





## 4 Discussion

Stability metrics are determined by the Reynolds decomposition window used to calculate them. Wind energy industry approaches have historically relied on a consistent 10 min averaging period, regardless of the stability regime (IEC; Bailey et al., 1997; Goit et al., 2022). This 10 min Reynolds decomposition window assumes (or potentially mis-assumes) clear separation between the large-scale mesoscale motions and the small-scale turbulent motions and that the 10 min Reynolds decomposition window appropriately distinguishes both regimes. In contrast, a longer, 30 min, Reynolds decomposition window has the potential to resolve more turbulent fluctuations while still avoiding larger-scale mesoscale motions. This difference is clearly seen in the maximum $TKE$ values during the middle of the day (Figs. 10, 11) where the 30 min window can include circulations that fill the deeper daytime convective boundary layer.

The Reynolds decomposition window impacts turbulence characterization on the SW ridge. Notably, 10 m $TKE$ differences between Reynolds decomposition windows are larger at the SW ridge (Fig. 10a) than in the valley (Fig. 10c) or on the NE ridge (Fig. 10e), both during stable and unstable periods. The same is true for the $u_*$ diurnal cycle (Fig. 8). Similarly, the $u_*$ ogive on the SW ridge reflects anomalous behavior during stable periods in which a 30 min averaging period is not sufficiently long (Fig. 3).

The Reynolds decomposition window also impacts stability characterization in the valley. The $L$ and $TKE$ partitions in the valley shift depending on whether a 30 min (Fig. 15a) or 10 min Reynolds decomposition window is used during stable periods (Fig. 15b, c). Similarly, the strength of $TKE$ stratification is also influenced by the Reynolds decomposition window.

One way to explain this apparent decoupling in the valley between the 10 m and 100 m measurements is to consider remote generation of turbulence that is then advected to the upper levels of the valley tower. This advection could occur horizontally, such that turbulence generated on the ridge is advected into the valley, where it is dissipated. This process could also occur vertically through the presence of an upside-down boundary layer (Parker and Raman, 1993; Mahrt, 1999). Warm air on the ridges may be advected over cold air pooled and trapped in the valley, leading to shear generation at the top of the cold pool (Mahrt, 1999). The explanation of an upside-down boundary layer in the valley is supported by a consistent $TKE$ increase across intermediate heights between 10 m and 100 m within the valley (Fig. 11c, d), but not at the two ridge locations (Fig. 11a, b, e, f).

Regardless of the location or the exact mechanism, the overall low HHPI values have potential implications for the utility of shorter towers when information at 100 m is needed. The low HHPI values (Fig. 13) suggest that surface measurements are unable to make hub-height predictions, regardless of the Reynolds decomposition window. Thus, meteorologists may not be able to avoid investing in hub-height-tall meteorological towers to make informed hub-height predictions.

The HoH results also have potential implications for those who need information over a broad area in complex terrain. Results from the Louvain community detection algorithm analysis of the observations suggest shared behavior among towers in common terrain (Fig. 15). Thus, complex terrain meteorologists may be able to reduce the number of towers necessary to understand surface stability by strategically placing tower locations in areas of differing terrain. Researchers and developers



may also consider implementing the Louvain community detection algorithm with flow models such as large-eddy simulations (Wise et al., 2021) to assess locations required for tower siting.

## 5 Conclusion

Characterizing atmospheric stability is crucial for multiple wind energy applications and becomes more challenging in heterogeneous complex terrain. This work improves atmospheric stability characterization in complex terrain by analyzing how atmospheric stability varies in the Perdigão valley. We assess the utility of building taller towers at three strategic locations and also consider the optimal tower number and placement to sample variability in the atmospheric stability for the complex site. We perform these assessments based on three stability metrics – the Obukhov Length ($L$), the turbulence kinetic energy ($TKE$), and the turbulence dissipation rate ($\epsilon$). We also repeat these assessments according to different Reynolds decomposition windows, considering a consistent 30 min averaging window common in the atmospheric science community and a consistent 10 min averaging that is standard in the wind energy community.

Results provide potentially valuable insights for those interested in analyzing areas of complex terrain. For example, 10 m surface measurements do not provide reliable 100 m hub-height stability predictions. Thus, it may be necessary to invest in meteorological towers that extend to hub height to make effective hub-height predictions.

Results here also provide insights regarding approaches to defining the number and location of meteorological towers necessary to accurately characterize a complex terrain site, at least in terms of atmospheric stability parameters such as $TKE$, $\epsilon$, and $L$. Different results make occur from variations in wind speed, wind direction, and/or turbulence intensity. Specifically, results from this analysis suggest that it may be possible to reduce the number of necessary meteorological towers by strategically sampling towers in regions of differing terrain and by coupling community detection algorithms with flow models like those in Wise et al. (2021).

Results also reflect how atmospheric stability characterizations are influenced by choices in the Reynolds decomposition window. The Reynolds decomposition window influences stability characterization in the valley, such that a 30 min Reynolds decomposition window during unstable periods better captures maximum values, and during stable periods clarifies the $TKE$ stratification in this location. Collectively, the results presented herein speak to the challenges that stakeholders and meteorologists may face in balancing the need for accurate characterizations of atmospheric stability in areas of complex terrain with the financial implications of investing in expensive meteorological towers.

These results also open up opportunities for future work that could inform improved characterization of atmospheric stability in complex terrain. For example, an application of the Louvain community detection algorithm to large-eddy simulations (Wise et al., 2021) for the Perdigão site could demonstrate the effectiveness of this approach for identifying critical locations for measurements. Such a tool could streamline future field experiment or wind resource assessment campaigns, or the design of mesonets (Brock et al., 1995; Brotzge et al., 2020).

Finally, future work may also involve extending the assessments presented in this analysis to other sites with complex terrain and with more stability metrics. For example, analyzing heavily vegetated sites and sites over water could improve



understanding of surface roughness length variability and the corresponding similarity relationships. Further, introducing more stability metrics could support stakeholders and meteorologists evaluating the relative value of investing in more sensors to calculate an Obukhov length as opposed to more cost-effective stability characterizations (Cantero et al., 2022).

*Code and data availability.* The Perdigão tower data can be found at: https://data.eol.ucar.edu/dataset/536.011 . The processing code can
be found at: [Insert upon acceptance]. The processed data can be found at:[Insert upon acceptance]. <A GitHub repository will be made available upon acceptance.>

*Author contributions.* JKL: conceptualization, funding acquisition, investigation, project administration, supervision, writing - original draft, writing - review & editing. NJA: formal analysis, investigation, methodology, software, validation, visualization, writing - original draft and writing - review & editing.

*Competing interests.* At least one of the (co-)authors is a member of the editorial board of *Wind Energy Science*.

*Acknowledgements.* We thank Ivana Stiperski and Samuele Mosso for discussions on Reynolds decomposition intervals. This research has been supported by the U.S. National Science Foundation (grant no. AGS-1565498 and CAREER AGS-1554055). This work utilized the Alpine high performance computing resource at the University of Colorado Boulder. Alpine is jointly funded by the University of Colorado Boulder, the University of Colorado Anschutz, Colorado State University, and the National Science Foundation (award 2201538). This work
was authored in part by the National Renewable Energy Laboratory for the U.S. Department of Energy (DOE) under Contract No. DE-AC36-08GO28308. Funding provided by U.S. Department of Energy Office of Energy Efficiency and Renewable Energy Wind Energy Technologies Office. The views expressed in the article do not necessarily represent the views of the DOE or the U.S. Government. The U.S. Government retains and the publisher, by accepting the article for publication, acknowledges that the U.S. Government retains a nonexclusive, paid-up, irrevocable, worldwide license to publish or reproduce the published form of this work, or allow others to do so, for U.S. Government
purposes.





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
