# Peer review of "Characterizing atmospheric stability in complex terrain"

_Wind Energy Science, 2025_

## Author Comment (AC1)

**We thank both the editors and reviewers for considering this manuscript.** Below, reviewer comments are in black and **our responses are in blue bold.**

**Reviewer 1**

This paper addresses an important and timely topic: characterizing atmospheric stability in complex terrain using Perdigao tower data. The authors evaluate stability metrics with different Reynolds decomposition intervals, and quantify the predictive skill of low level vs hub height stability. They apply clustering methods to recommend a necessary number of met masts which is good but rather academic because usually the problem is the opposite: how many more met masts than 1 are needed to properly sample the site conditions (and where numerical estimates of wind field covariance are helpful - see the other general comment).

The analysis is carefully executed, the dataset is of very high quality, and the topic is relevant to both atmospheric science and wind energy.

**We thank the reviewer for their time and careful consideration of our work.**

However, the paper in its current form is incomplete. Perdigao has also been extensively studied with mesoscale and large-eddy simulation (LES) modeling, yet the study relies exclusively on observational/statistical analyses, while ignoring the context and assistance that numerical models can provide. Without at least a discussion — and preferably some demonstration — **of how models complement observations**, the results risk being overly narrow and less generalizable, especially when the conclusions are drawn from data from a single site.

**We thank the reviewer for this thoughtful perspective. We also welcome the opportunity to demonstrate how our proposed Louvain methodology can be extended to model data. As such, we have introduced another analysis of the Louvain community detection algorithm that relies on the publicly-available LES analysis of the Perdigão valley performed in Robey and Lundquist (2024). We hope that this established case study, as well as its corresponding analysis, demonstrate one potential application of our proposed methodology as well as motivate future research. This work can be found in the new Appendix C:**

**Here we demonstrate a proof-of-concept implementation of the Louvain methodology applied to large-eddy simulation (LES) data. This Louvain implementation considers near-surface static stability from the LES of Robey and Lundquist (2024) (Table C1), based on Wise et al. (2021). These data represent a roughly 3 hr period early in the morning. Although simulation output is available at 1 second resolution, here we sample every 30 min, consistent with other analyses in this work. We also spatially coarsened the 100 m output to the effective model resolution of 500 m (Skamarock,**

2004) and then subset to the identified Perdigão tower locations. Static stability was then calculated from these spatially- and temporally-coarsened data as:

$$\frac{\partial \theta}{\partial z} = \frac{\theta_2 - \theta_1}{z_2 - z_1}$$

such that θ represents a potential temperature (K), z represents a model height above ground (m) and the two model levels taken are those closest to the surface (10 m and 2 m). The Louvain algorithm was then applied to these static stability calculations, resulting in seven partitions (Fig. C1). Overall, these tower groupings reflect broad terrain boundaries, with opportunities for mixing. Orange is predominately restricted to the upper valley. Dark blue exists mostly on ridges and the outer transects. The relatively small gray partition exists in the slope between the NE ridge and the valley. Light blue reflects most of the SW ridge, with occasional representation in the valley. The salmon, brown, and green transects are less geographically informed. Of course, because this analysis is based on 3 hours of LES during stably stratified conditions rather than the full measurement campaign period, the partitions are different from those presented in the main paper.

**Table A1. Model parameters used in LES Louvain analysis, based on Robey and Lundquist (2024).**

| Model Parameter | Value |
|---|---|
| Start time | 2017-06-14 03:24:00 |
| End time | 2017-06-14 05:16:25 |
| Effective horizontal resolution | 500 m |
| Horizontal grid resolution | 100 m |
| Vertical grid resolution | 160 vertical levels |
| Vertical levels | 2 m and 10 m |
| Land cover | CORINE Land Cover 2006 with mixed shrubland–grassland roughness length updated to 0.5 m |
| Terrain | 1 arcsec terrain from the Shuttle Radar Topography Mission (SRTM) |
| Turbulence subgrid-scale model | non-linear backscatter and anistropy (NBA2) |

[Figure]

**Figure A1. Static stability partitions determined by the Louvain community detection algorithm with LES output from Robey and Lundquist (2024) at the locations of the station towers. Model parameters are defined in Table A1.**

**"This implementation is designed to serve several stakeholders and future research directions. Because this analysis builds on publicly available LES output from Robey and Lundquist (2024), it provides a reproducible benchmark for future Louvain or LES-based studies. The short duration and dense observations create opportunities to explore similarity metrics beyond the temporal correlation weight used here. We also introduce static stability as an alternate metric that depends only on temperature measurements, recognizing that many stakeholders, including wind farm developers, rely on sparse instrumentation. To further mimic these practical constraints, we adopt a 30-minute temporal resolution.**

**"This case study therefore provides both a benchmark and a flexible framework that can be extended to other datasets, parameterizations, and applications."**

I therefore recommend major revision to address the points above, and a few specific comments below.

Specific comments

L32: Discussing various papers about the effects of stability before at least defining it broadly. Some of these papers even analyze data from complex terrain.

**We thank the reviewer for this comment. We agree that beginning more broadly helps contextualize the scope. To avoid describing the effects of something that is not yet defined, we propose beginning by contextualizing atmospheric stability with respect to its connection to atmospheric boundary layer as a whole:**

**"The atmospheric boundary layer (ABL) directly interacts with Earth's surface, and the surface/atmosphere interactions that occur in this layer dictate transport of heat, momentum, sediment, and moisture into the larger atmosphere (Stull, 1988). Understanding these exchange processes has implications for fields like renewable energy (Pérez Albornoz et al., 2022) and agriculture (Tang et al., 2022). These exchange processes are also highly dynamic and sensitive to atmospheric stability (Garratt, 1994)."**

L152: The Obukhov length is not proportional to the height above the surface. It is the height above the surface.

**Thank you. We adapt this line, now line 210, to properly read:**

**"The Obukhov Length, *L*, is the height above the surface…"**

L162: Please define Tv. Is it even meaningful to use theta-v and Tv in the context when theta is then anyway assumed to be constant?

**We appreciate the reviewer's thoughtful observation regarding the differences in temperature characterization. We agree that moisture and pressure corrections to air temperatures could allow for fine adjustments to the analysis. We also believe that the most appropriate way to respond to the available measurements is to name the experimental limitation as opposed to either misrepresenting the theory or introducing proxies that further propagate uncertainties. To improve clarity, this line, now line 220, reads:**

**"where $T_v$ is the virtual temperature (K) approximated as the air temperature; $p_0$ is a reference pressure (Pa); $p$ is the pressure at a given height (Pa); and $\frac{R}{C_p} \approx 0.286$.**

**Herein, $\theta_v$ was approximated as the air temperature due to a lack of localized**

**pressure and moisture measurements. Further, because not all 47 towers had temperature measurements available at 10 m, the virtual potential temperature for the horizontal homogeneity analysis (described later) was assumed to be a uniform 300 K."**

L250: Please clarify if the linear regression is calculated at specific time-stamps? If yes, please discuss how the vertical information propagation may adversely affect this metrics.

**We thank the reviewer for this comment. The linear regression incorporates all 30 min (or 10 min) timesteps. Thus, while decoupling occurs for 8-11% of the (cloudless) field campaign, the vertical information propagation over the 100-m tower is also significantly shorter than the Reynolds averaging window. We acknowledge the concern on line 311:**

**"The vertical information propagation over the 100-m tower is also assumed to be much shorter than the Reynolds decomposition window."**

L274, Figure 2: The case (e) tse09 (valley) stable exhibits the opposite behavior from every other case. This would merit some discussion.

**We thank the reviewer for this observation. We agree that the stable conditions in the valley are notable and worthy of discussion. Discussion of this behavior, in part, has informed the development of a new section of the manuscript, 3.1.3 Vertical Profiles and a new figure, Fig. 6, all of which highlight the unique behavior of the stable profiles in the valley.**

[Figure]

**Figure 6. Perdigão field campaign vertical profiles with 30 min averaging at the three 100 m towers. The solid line indicates the mean value and the band represents the standard error. (a) friction velocity all (00-23 UTC); (b) friction velocity stable (00-02 UTC); (c) friction velocity unstable (12-14 UTC); (d) heat flux all (00-23 UTC); (e) heat flux stable (00-02 UTC); (f) heat flux unstable (12-14 UTC).**

**The dedicated discussion of the stable heat flux profiles in the valley begins on line 418:**

**"Stable heat flux profiles also reflect profoundly distinct shapes between the ridges and the valley (Fig. 6e). Stable heat flux profiles along the two ridges are more stable than in the valley at the surface, but as the height increases, the stable heat fluxes along the two ridges become increasingly weaker (Fig. 6e). In contrast, the stable heat flux profile in the valley (Fig. 6e) becomes more stable from the surface to 40 m before stabilizing above. Thus, while the valley is well-mixed during stable conditions, the ridges are not (Fig. 6e). This unique heat flux behavior in the valley during stable conditions is also corroborated by the heat flux CFAW-CM analysis, where stable heat fluxes in the valley (Fig. 4e) are much tighter and smaller than those along the two ridges (Fig. 4b,h). These ridge/valley differences in stable heat fluxes may signal an upside down boundary layer in the valley (Fig. 6e)."**

**We also complement this discussion with another, detailed discussion of an upside down boundary layer in the valley later on, starting on line 503:**

**"The TKE in the valley (Fig. 14c, d) shows stratified layers during nighttime hours, with smaller values at the surface and larger values with increasing height.**

**"One way to explain this apparent decoupling in the valley between the 10 m and 100 m measurements is to consider remote generation of turbulence that is then advected to the upper levels of the valley tower. This advection could occur horizontally, such that turbulence generated on the ridge is advected into the valley, where it is dissipated. This process could also occur vertically through the presence of an upside-down boundary layer (Parker and Raman, 1993; Mahrt, 1999). Warm air on the ridges may be advected over cold air pooled and trapped in the valley, leading to shear generation at the top of the cold pool (Mahrt, 1999). The explanation of an upside-down boundary layer in the valley is supported by a consistent TKE increase across intermediate heights between 10 m and 100 m within the valley (Fig. 14c, d), but not at the two ridge locations (Fig. 14a,b, e, f). This unique, well-mixed behavior during stable conditions in the valley is also reflected in the heat fluxes (Fig. 6e)."**

L277: Not entirely clear how it is discernible from Fig. 2h that the stable ogive shifts to mesoscale fluctuations at 60 min. Do you mean that the curves which appear to have flattened, suddenly receive a kink?

**We thank the reviewer for the opportunity to be more clear in our descriptions. The reviewer is correct that we are in fact commenting on the sudden shift/kink that occurs after the otherwise flattened behavior. We adapt this line, now line 399, to read:**

**"On the NE ridge, heat fluxes (Fig. 4h) are mostly constant through 30 min before increasing from 30-60 min."**

L307: "more diffuse" is not the best choice of words. It would be better said that the winds are less bidirectional, or aligned with the terrain.

**We thank the reviewer for this comment. We alter this line (now line 448) to read "less bidirectional."**

L310, Figure 3: Please discuss why the ogives (are they really ogives, strictly speaking?) for u* are so different from those for the heat flux?

**Thank you for both the motivation for discussing u\* vs. heat flux as well as challenging our labelling of our method as an ogive method.**

**Regarding the first issue, we argue that the friction velocity ogives differ from those for heat flux because of the increased surface roughness from surface and canopy cover.**

**Regarding the second issue, whether these are strictly speaking ogives, we have carefully considered the differences between our analysis in the time domain and traditional ogive analysis in the frequency domain. While these approaches should result in the same conclusion, the fact that we did our analysis in the time domain does suggest that we should label the approach with a different name. Therefore, we have labelled our approach the "Cumulative Flux Averaging-Window Convergence Method (CFAW-CM)" and distinguished it from the ogive method in the text as below:**

**"An appropriate averaging window was determined using an ogive-type method, the Cumulative Flux Averaging-Window Convergence Method (CFAW-CM). While the ogive method (Desjardins et al., 1989; Oncley et al., 1996; Babic et al., 2012) identifies an appropriate Reynolds-averaging window through spectral integration of the flux cospectrum in the frequency domain, the convergence method determines the window in the time domain by identifying the averaging period beyond which additional low-frequency variability no longer contributes to the flux covariance. This assessment was performed to determine whether a uniform averaging period could be utilized for heat fluxes ($\overline{w'\theta'_{v}}$) and friction velocities ($\overline{u_{*}}$) across various heights, times of day, and tower locations.**

**"By evaluating the cumulative eddy flux contribution according to increasing averaging periods, an asymptote separates the local turbulent fluctuations from the larger-scale (mesoscale) fluctuations. For this dataset, heat flux and friction velocity values were calculated at each height for each tower location according to 1, 5, 10, 20, 30, and 60 min Reynolds decomposition windows.**

L477: It is too optimistic to claim that this work improves the characterization of atmospheric stability. It only demonstrates the challenges, based on data from one site.

**We thank the reviewer for this comment. We alter new line 626 to read "This work demonstrates the challenges in atmospheric stability characterization in complex terrain…"**

L486: "... towers that extend to hub height ...". Should they not be extended to the rotor top?

Arguably, similar if not larger discrepancies will occur when the top of the boundary layer, which often is at about HH, intersects the rotor area.

**We thank the reviewer for this comment. We also agree that the rotor top may present an additional transition in stability characterization. However, because measurements are not available at the rotor top for this analysis, we restrict our conclusions to a designated 100 m hub-height. As such, we propose to keep the following line, now line 634, to read:**

**"Thus, it may be necessary to invest in meteorological towers that extend to hub height to make effective hub-height predictions."**

**We also clarify this decision in the text starting on line 309:**

**"While many representative heights may be potentially considered in a wind farm resource assessment, the hub height is one such representative measurement. Further, the hub height, with typical values near 100 m, aligns with the highest available measurement from the meteorological towers during the Perdigão field campaign."**

---

## Author Comment (AC2)

**We thank both the editors and reviewers for considering this manuscript.** Below, reviewer comments are in black and **our responses are in blue bold.**

**Reviewer 2**

Review of Characterizing atmospheric stability in complex terrain by Nathan J. Agarwal and Julie K. Lundquist

Recommendation: Major revision

This is a well written, careful study that leverages the unusually rich Perdigão tower dataset to assess how three stability metrics (Obukhov length L, turbulence kinetic energy TKE, and dissipation rate $\epsilon$) behave in complex ridge–valley terrain and how representative surface (10 m) measurements are of hub-height (100 m) conditions. The manuscript provides useful guidance on Reynolds-decomposition windows (ogive analysis) and introduces sensible metrics for quantifying representativeness (HHPI and horizontal homogeneity). Overall the work is scientifically sound and valuable to the wind-energy and boundary-layer communities, but several important methodological clarifications and additional discussion (especially on canopy effects and comparisons to flat-terrain benchmarks) are required before publication.

**We thank the reviewer for their thoughtful and supportive appraisal.**

Major comments

1.  Expand and tighten the Introduction with regard to prior Perdigão work.

o    The Perdigão campaign generated many multi-instrument studies on ridge–valley and canopy effects. Readers would benefit from a clearer map of what previous Perdigão studies (and other complex-terrain works) have shown and how this manuscript's contribution differs.

**We thank the reviewer for this comment. We agree that the Perdigão field campaign has allowed for several laudable scientific contributions. We include a more complete peer review that begins on (updated) line 109 and extends fully to line 144, in section 2.1:**

**"Perdigão addressed fundamental understandings of physical mechanisms in complex terrain. Menke et al. (2019) documented recirculation zones throughout the valley with doppler radar data. Letson et al. (2019) used data from both sonic anemometers and Doppler lidars to assess wind gust characteristics. Wagner et al.**

(2019) as well as Wildmann et al. (2019) and Venkatraman et al. (2023) analyzed detection of low-level jets (LLJs). Wise et al. (2021) and Robey and Lundquist (2024) have documented mountain wave patterns. This field campaign has also supported research into wind-energy-specific mechanisms like wakes. Menke et al. (2018) used data from six scanning lidars to analyze the dependence of wake deflection patterns on the atmospheric stability, and these wake deflection patterns were also scrutinized by Barthelmie and Pryor (2019) and Wise et al. (2021). Similarly, Dar et al. (2019) simulated the wake behind a turbine and showed that the self-similarity of the wake implied that the wake pattern depended on neither the complexity nor the shape of the terrain.

"Improving the understanding of these physical mechanisms has also enabled model improvement. Wagner et al. (2019) established a foundation with a long-term Weather Research and Forecasting (WRF) large eddy simulation (LES) simulation set for the full observational period. Palma et al. (2020) developed a digital terrain model (DTM) for the site and also defined a necessary horizontal grid cell spacing criteria of 40 m to appropriately characterize the site. Several modeling works have also expanded upon this need for improved representation of horizontal variability for the region. Quimbayo-Duarte et al. (2022) established a forest canopy parameterization for the WRF-LES model that improved wind modeling within the lowest 500 m of the atmosphere and attributed the imposed drag parameterization to a need for even higher-resolution land surface inputs. Venkatraman et al. (2023) altered the forest canopy parameterization to consider more realistic (i.e. shorter) heights in a set of OpenFOAM simulations. Finally, Al Oqaily et al. (2025)—building upon the ensemble models in Giani and Crippa (2024)—again underscored the importance of land cover in an additional set of LES simulations. Connolly et al. (2021) demonstrated the utility of the cell perturbation method (CPM) (Muñoz-Esparza et al., 2015) in capturing flow features in LES simulations of the region, especially in weakly convective conditions. The Perdigão field campaign has also enabled improvements in wind turbine specific modeling in complex terrain. Wise et al. (2021) introduced the WRF-LES-GAD to the Perdigão dataset, and leveraging the output from this model, Robey and Lundquist (2024) demonstrated the utility of a virtual lidar model (Robey and Lundquist, 2022) in simulating range height indicator (RHI) scans. The modeling improvements have also extended beyond numerical weather prediction (NWP) models. Barthelmie and Pryor (2019) developed a novel detection algorithm that could both identify and characterize wake patterns, Vassallo et al. (2020) used artificial neural networks (ANNs) to improve wind speed error by up to 52%, Bodini et al. (2020) used machine-learning models to reduce the average error in model representation of turbulence dissipation rate by up to 40%, and Mosso et al. (2025) predicted turbulence anisotropy for the region using random forest models.

"The Perdigão field campaign has also enabled improvements in measurement techniques. Vasiljevic et al. (2017) outlined a methodology for multi-lidar Doppler

**lidar experiments and Wildmann et al. (2019) presented a new wake measurement strategy that used three synchronized lidars to adapt to the prevailing wind direction automatically. Bell et al. (2020) combined multiple Doppler lidar scans to create virtual towers that could extend beyond the range of traditional in situ meteorological towers. Coimbra et al. (2025) also performed a virtual mast analysis of the Perdigão region with three coordinated dual lidar. These measurements allowed for not only the mean quantity comparisons available in Bell et al. (2020), but also turbulent quantities as well."**

**We also welcome an opportunity to more clearly motivate the novelty of the present work. We include an additional paragraph in the Introduction starting on line 91:**

**"(E)xisting Perdigão analyses share two important limitations. First, most focus on vertical extrapolation, virtual towers, and rotor-layer prediction while largely neglecting how horizontal heterogeneity imposed by steep terrain and vegetation modulates near-surface flow. Second, although technological solutions have enabled reliance on measurements above 40–80 m, the representativeness of surface or 10 m observations remains poorly understood. This gap directly affects wind-energy stakeholders, who often depend on near-surface measurements for resource assessment and may be unable to deploy tall towers in complex terrain.**

**"These limitations motivate the present study…"**

o    Menke et al. (2019; Characterization of flow recirculation zones at Perdigão), a study co-authored by one of the present authors, also investigates stability effects and should be discussed — particularly in the context of how recirculation zones may provide alternative explanations for some of the vertical decoupling observed here and why the stability characterization approach used in Menke et al. is not used in this work. If the authors consider Menke et al. (2019) not directly comparable, they should explicitly state and justify this.

**We thank the reviewer for this comment. We agree that the Menke et al. (2019) work is an important contribution to the Perdigão literature and include it in the literature review. We avoided using a Richardson number because, as our analysis demonstrates, the stability at one level may not necessarily reflect the stability at another level. To reflect this concern in the manuscript, we have added a line, now line 206:**

**"Richardson Numbers were also intentionally not employed for this analysis. While other analyses such as Menke et al. (2019) have relied on a gradient Richardson number to assess stability, here we avoid any layer-based stability characterizations. These metrics were excluded because these metrics are sensitive to how the layer is defined."**

We also understand the reviewer's interest in a discussion as to the potential role of recirculation zones in contributing to the vertical decoupling.

We also address TKE decoupling directly on line 521:

"Canopy effects may help explain this 10 m TKE decoupling during unstable conditions in the valley and NE ridge. These 10 m measurements, located with the canopy layer but not necessarily the vegetation layer, could experience heightened turbulence that would necessarily be reflected at taller measurements. Recirculation zones, also present during unstable conditions in the valley and NE ridge, may also influence this TKE decoupling. However, as noted in Menke et al. (2019), the depth of these recirculation zones exceeds 100 m. Thus, because these recirculation zones would not necessarily impact 10 m measurements uniquely, their effect may be secondary to the more direct influence of the canopy."

We also discuss the potential role of recirculation zones in contributing to the observed friction velocity behavior on the NE ridge during unstable conditions. On line 431 in a new Vertical Profile section, where we discuss how–during unstable conditions, NE ridge friction velocities align more with valley friction velocities than SW ridge friction velocities–we note:

"Both flow and terrain features may be contributing to this NE ridge subversion. Menke et al. (2019) documented recirculation zones during over half of the unstable periods. These flows could then explain why this unique behavior is reflected only during unstable cases. Menke et al. (2019) also documented that these recirculation flows occurred more frequently near the valley and NE ridge than the SW ridge. Thus, recirculation zones provide a mechanism that is aligned in terms of geography and stability condition to explain this behavior..."

[Figure]

**Figure 6. Perdigão field campaign vertical profiles with 30 min averaging at the three 100 m towers. The solid line indicates the mean value and the band represents the standard error. (a) friction velocity all (00-23 UTC); (b) friction velocity stable (00-02 UTC); (c) friction velocity unstable (12-14 UTC); (d) heat flux all (00-23 UTC); (e) heat flux stable (00-02 UTC); (f) heat flux unstable (12-14 UTC).**

2.   Significantly expand discussion and quantification of canopy/vegetation effects.

o   Why it matters: Several of the manuscript's central findings — in particular the apparent 10 m vs 100 m decoupling in the valley and the NE ridge — could be strongly influenced by canopy height and density. The manuscript repeatedly notes "canopy" or "canopied NE ridge" but does not quantify canopy height/cover or show photos/maps of tower surroundings. The text suggests remote generation/advection as the explanation (e.g., upside-down boundary layer / advection), but canopy sheltering is an equally plausible (and in places more parsimonious) explanation for decoupling at 10 m. See the discussion and TKE stratification text and figures.

o   Suggested fixes:

§ Add a table or figure that lists canopy/land-cover at each tower (or at least for the three focal towers), **including approximate canopy height and whether the 10 m sonic is inside the roughness sublayer/canopy.**

**We thank the reviewer for this comment. We include a new table and associated text (starting on line 156) with the requested information. We base our estimates on existing literature for this case study.**

**"The three 100 m towers were strategically placed to sample flow conditions within differing terrains. Specifically, tse04 was located on an exposed ridge on the southwest, tse09 in the valley with eucalyptus and fir trees, and tse13 on a ridge with a heterogeneous canopy to the northeast (Table 1). Although translating CORINE Land Cover data (Bossard et al., 2000) into U.S. Geological Survey land use types to obtain surface roughness lengths (Pineda et al., 2004) suggests surface roughness lengths on the order of 0.01–0.05 m, Wise et al. (2021), Wagner et al. (2019), and Palma et al. (2020) (among many others) have suggested that larger surface roughness lengths are more appropriate for the site. The mean canopy heights–as documented in Letson et al. (2019)–are below 10 m (Table 1 and Fig. 2), although individual trees may extend to 15 m (Mosso et al., 2025)."**

**Table 1.** Shortname Reference for 100 m towers

| Tower | Shortname | Topography | Location [WGS84] | Elevation above sea level [m] | Mean canopy height [m] |
|-------|-----------|------------|------------------|-------------------------------|------------------------|
| tse04 | Southwest Ridge | Turbine Ridge | 7°44′33.37″W 39°42′21.47″N | 473 | 1.5 |
| tse09 | Valley | Valley | 7°44′5.40″W 39°42′40.36.″N | 305 | 5.4 |
| tse13 | Northeast Ridge | Canopy Ridge | 7°43′49.38″W 39°42′48.97.″N | 453 | 3.6 |

§ Include **site photographs** of the three focal towers (or reference an existing photo repository) to make interpretation of 10 m measurements transparent to readers.

**"We thank the reviewer for this suggestion. We include the below figure, Fig. 2, which includes available site photographs of each of the three focal towers.**

[Figure]

[Figure]

[Figure]

**Figure 2. Canopy cover for (a) tse04, (b) tse09, and (c) tse13. The image of tse04 was identified as a still photo in the Danish Technical University (DTU) Perdigão video (DTU Wind and Energy Systems, 2017) and the images of tse09 and tse13 can be found in the Perdigão data archive (EOL).**

§ Re-analyze (or at least discuss) whether the valley / SE ridge 10 m decoupling is better explained by canopy effects than by remote advection.

**We thank the reviewer for their proposal of an alternate decoupling mechanism. We include a more complete discussion of TKE decoupling mechanisms, emphasizing**

**that remote advection may likely be the main driver during stable conditions, while canopy effects may become more prominent during unstable conditions.**

**Starting on line 506:**

**"One way to explain this apparent decoupling in the valley between the 10 m and 100 m measurements is to consider remote generation of turbulence that is then advected to the upper levels of the valley tower. This advection could occur horizontally, such that turbulence generated on the ridge is advected into the valley, where it is dissipated. This process could also occur vertically through the presence of an upside-down boundary layer (Parker and Raman, 1993; Mahrt, 1999). Warm air on the ridges may be advected over cold air pooled and trapped in the valley, leading to shear generation at the top of the cold pool (Mahrt, 1999). The explanation of an upside-down boundary layer in the valley is supported by a consistent TKE increase across intermediate heights between 10 m and 100 m within the valley (Fig. 14c, d), but not at the two ridge locations (Fig. 14a, b, e, f)."**

**And again at line 523:**

**"Canopy effects may help explain this 10 m TKE decoupling during unstable conditions in the valley and NE ridge. These 10 m measurements, located within the canopy layer but not necessarily within the vegetation layer, could experience heightened turbulence that would necessarily be reflected at taller measurements. Recirculation zones, also present during unstable conditions in the valley and NE ridge, may also influence this T KE decoupling. However, as noted in Menke et al. (2019), the depth of these recirculation zones exceeds 100 m. Thus, because these recirculation zones would not necessarily impact 10 m measurements uniquely, their effect may be secondary to the more direct influence of the canopy.**

3.   Justify the choice to focus horizontal-homogeneity (HoH) on 10 m measurements only, or extend analysis to higher levels.

o   The HoH analysis determines the minimum number of towers needed to represent site variability but is performed at 10 m because only 18 towers have 10 m temperature. For wind-energy stakeholders, representativeness at hub-height (or intermediate heights) is more relevant than representativeness at 10 m.

**We thank the reviewer and also wish to clarify our methodological choice here. The temperature measurement availability was not the motivating factor for performing our analysis at 10 m. Instead, we determined, a priori, to perform the analysis at 10 m to explore opportunities to leverage surface characterizations that wind farm developers routinely use in early siting. By using 10-m towers, we are able to use all 47 towers. Other altitudes would provide fewer data points for consideration. Shifting**

**to 20 m would only allow 38 towers to be analyzed. Shifting to 30 m would only leave 17 towers. Further, only 3 towers had measurements at hub height.**

o   Suggested fixes:

§   Either: (a) repeat the HoH analysis at other heights where sufficient data exist (e.g., 20/40/60 m subsets) and show differences, or **(b) clearly justify why a 10 m-based HoH is still useful for turbine siting.**

**We thank the reviewer for this suggestion. While including measurements from taller heights may seem ideal, in this particular case, shifting the analysis to a height above 10 m reduces available data and thus the impact of the HoH analysis. Shifting to 20 m would only allow 38 towers to be analyzed. Shifting to 30 m would only leave 17 towers. Analyzing 100 m would only leave 3 data points. Further, 10-m towers are much more common in wind resource assessment than any of the other heights. Maintaining the analysis at 10 m also reveals that stakeholders may still be able to leverage the commonly-used 10-m towers to inform site characterization. To acknowledge this methodological concern in the manuscript, we have added additional text in a paragraph, now starting on line 319:**

**"Each metric was also evaluated for HoH. This assessment was designed to quantify the smallest number and location of meteorological towers necessary to capture the site's variability in surface (10 m) stability. For each metric, all 47 towers with sonic anemometers were considered. Because only 18 of these towers had available 10 m temperature measurements, the virtual potential temperature was assumed to be a uniform 300 K for the Obukhov Length calculation just for the HoH analysis. Thus, this analysis could not account for differences in virtual potential temperature that might occur between towers. This approach to addressing the lack of available temperature measurements was determined optimal because other potential solutions–such as performing the analysis at either a shorter or taller height–further limited the number of available towers or relied on data fully within the vegetation layer. Shifting to 20 m would only allow 38 of the 47 towers to be analyzed. Shifting to 30 m would only leave 17 towers. Analyzing 100 m would only leave 3 data points. These other solutions also devalued the assessment as a practical demonstration of how stakeholders could leverage surface-level measurements to inform site characterization."**

§   Add a short paragraph discussing pros/cons of using Louvain community detection (e.g., sensitivity to input correlation metric and partition resolution) and whether other clustering algorithms were tested.

**We thank the reviewer for this careful observation. Our decision to use the Louvain community detection algorithm is largely because this algorithm is most appropriate for the use case. Other clustering algorithms, such as a kNN, require the analyst to pre-determine the number of clusters. As such, the kNN approach ambiguously defines the marginal value of an additional cluster. To reflect this decision, along with further details on some of the benefits and limitations of this algorithm, we revise this paragraph starting on line 329:**

**"These 47 towers were then represented as a graph network, with a node for each tower location and the edge weights representing the strength of similarity in surface stability between any two given towers. This similarity metric S mapped the correlation in surface stability Cij with only positive edge weights such that S = (Cij +1)/2 . The smallest number of meteorological towers were then determined based on the Louvain community detection algorithm (Blondel et al., 2008), as implemented by the NetworkX Python package (Hagberg et al., 2008). Note that the Louvain algorithm, unlike kNN methods, treats the number of clusters as an output instead of an input. The Louvain algorithm determines which towers are redundant by identifying tower subgroups and assigning each tower to one of these subgroups. The Louvain community detection algorithm also enforces (non)treatment of overlapping communities, further ensuring that each community is treated independently with a unique tower. While the Louvain community detection algorithm is also sensitive to several tuning parameters, we make no adjustments to the default modularity gain threshold (1e-7) or resolution (1) and introduce no artificial seeds or establish restrictions on the number of optimization cycles."**

§    Consider adding the widely used TRIX (or equivalent) representativeness test from the wind-energy community to complement the HoH analysis — readers from industry will find that comparison useful.

**We appreciate the reviewer's suggestion to introduce a model-based HoH analysis to complement the observational analysis. We agree that a model-based HoH is a more complete demonstration of our methodology. While we did consider the TRIX representativeness test, we instead offer an LES-based analysis, consistent with the recommendation from Reviewer 1. We believe that an analysis of LES data provides a more natural comparison with our current observational analysis. This analysis is described in Appendix C:**

**"Here we demonstrate a proof-of-concept implementation of the Louvain methodology applied to large-eddy simulation (LES) data. This Louvain implementation considers near-surface static stability from the LES of Robey and Lundquist (2024) (Table A1), based on Wise et al. (2021). These data represent a roughly 3 hr period early in the morning. Although simulation output is available at 1**

second resolution, here we sample every 30 min, consistent with other analyses in this work. We also spatially coarsened the 100 m output to the effective model resolution of 500 m (Skamarock, 2004) and then subset to the identified Perdigão tower locations. Static stability was then calculated from these spatially- and temporally-coarsened data as:

$$\frac{\partial \theta}{\partial z} = \frac{\theta_2 - \theta_1}{z_2 - z_1}$$

such that θ represents a potential temperature (K), z represents a model height above ground (m) and the two model levels taken are those closest to the surface (10 m and 2 m). The Louvain algorithm was then applied to these static stability calculations, resulting in seven partitions (Fig. C1). Overall, these tower groupings reflect broad terrain boundaries, with opportunities for mixing. Orange is predominately restricted to the upper valley. Dark blue exists mostly on ridges and the outer transects. The relatively small gray partition exists in the slope between the NE ridge and the valley. Light blue reflects most of the SW ridge, with occasional representation in the valley. The salmon, brown, and green transects are less geographically informed. Of course, because this analysis is based on 3 hours of LES during stably stratified conditions rather than the full measurement campaign period, the partitions are different from those presented in the main paper.

Table A1. Model parameters used in LES Louvain analysis, based on Robey and Lundquist (2024).

| Model Parameter | Value |
|---|---|
| Start time | 2017-06-14 03:24:00 |
| End time | 2017-06-14 05:16:25 |
| Effective horizontal resolution | 500 m |
| Horizontal grid resolution | 100 m |
| Vertical grid resolution | 160 vertical levels |
| Vertical levels | 2 m and 10 m |
| Land cover | CORINE Land Cover 2006 with mixed shrubland–grassland roughness length updated to 0.5 m |
| Terrain | 1 arcsec terrain from the Shuttle Radar Topography Mission (SRTM) |
| Turbulence subgrid-scale model | non-linear backscatter and anistropy (NBA2) |

[Figure]

**Figure A1. Static stability partitions determined by the Louvain community detection algorithm with LES output from Robey and Lundquist (2024) at the locations of the station towers. Model parameters are defined in Table A1.**

**"This implementation is designed to serve several stakeholders and future research directions. Because this analysis builds on publicly available LES output from Robey and Lundquist (2024), it provides a reproducible benchmark for future Louvain or LES-based studies. The short duration and dense observations create opportunities to explore similarity metrics beyond the temporal correlation weight used here. We also introduce static stability as an alternate metric that depends only on temperature measurements, recognizing that many stakeholders, including wind farm developers, rely on sparse instrumentation. To further mimic these practical constraints, we adopt a 30-minute temporal resolution.**

**"This case study therefore provides both a benchmark and a flexible framework that can be extended to other datasets, parameterizations, and applications."**

4. Provide a direct comparison or benchmark of HHPI with flat-terrain results (or literature).

o   The conclusion that "surface measurements are unable to make hub-height predictions" is important, but the reader needs context: are the HHPI values unusually low compared to flat terrain or other published sites? That context would quantify how "bad" the Perdigão complex terrain results are and support the recommendation for tall towers. The manuscript suggests future work comparing to other sites but does not provide **literature benchmarks**.

**We thank the reviewer for this suggestion. After assessing the available literature to confirm that our HHPI methodology is novel, we established our own HHPI benchmark with available eXperimental Planetary boundary layer Instrumentation Assessment (XPIA) data as described in Appendix B:**

**"Here we provide a flat terrain comparison to the Perdigão HHPI analysis. This flat terrain benchmark serves to contextualize the role of complex terrain in affecting the relative agreement between surface and hub-height stability. Sonic anemometer data from the Boulder Atmospheric Observatory (BAO) tower during the three-month eXperimental Planetary boundary layer Instrumentation Assessment (XPIA) (Lundquist et al., 2017; Bodini et al., 2018) field campaign provide the data source for this comparison. Available measurements include data from 5m, 50m, and 100m. As such, we compare both the 5 m - 100 m and 50 m - 100 m. Each stability metric calculation employs a screening process similar to that employed for the Perdigão data, including removal of the approximately 20% of data affected by precipitation, 99.5th percentile extrema removal, and tower wake distortion screening.**

**"HHPI is higher in flat terrain than in complex terrain at shorter heights. HHPI in flat terrain is similar to HHPI in complex terrain in several ways. Both improve with taller towers. In both cases, log10ϵ has the lowest HHPI of all metrics and this metric distinction is especially-pronounced for shorter heights (Fig. B1c). Both terrains also exhibit similar curvatures for each metric. L HHPI consistently is flatter at lower heights, and increases more sharply at higher heights (Fig. B1a). TKE HHPI increases more sharply for shorter heights before increasing more gradually above (Fig. B1b). log10 ϵ HHPI is unique in its consistent linear increase (Fig. B1). HHPI in the two terrains also differ. Notably, HHPI is higher in flat terrain than in complex terrain (Fig. B1). Surface L HHPI is between 0.1 and 0.2 higher in flat terrain than in complex terrain (Fig. B1a). Surface TKE HHPI in flat terrain leads that in complex terrain by sometimes as much as 0.3 (Fig. B1b). Surface log10 ϵ demonstrates the smallest flat terrain lead (Fig. B1c). These flat terrain leads, pronounced at the surface, are not guaranteed at higher heights. While L HHPI continues to reflect a flat terrain premium (Fig. B1a), both TKE HHPI (Fig. B1b) and log10 ϵ HHPI (Fig. B1c) do not. Thus, while shorter towers may be more useful in flat terrain than in complex terrain, flat terrain towers may also benefit less from height increases."**

[Figure]

**Figure B1. Flat terrain (XPIA) benchmark to the Perdigão HHPI analysis. (a) L; (b) TKE; (c) log10ϵ.**

5.    Re-evaluate some interpretations of Figures (wind roses and 10 m vs 100 m diffuseness).

o    The manuscript states "100 m winds are more diffuse than 10 m"; however, the reviewer's read of Fig. 4 suggests 100 m sometimes shows a clearer prevailing direction than 10 m. Please verify the figure interpretation and ensure the text statement is correct.

**We revise this sentence, now on line 448, to read "100 m winds are less bidirectional than 10 m winds" as suggested by another reviewer.**

Minor comments and editorial suggestions

1.    Introduction: Many studies cited that are not explicitly identified as flat or complex-terrain studies. For each "stability affects wind generation" citation, briefly state terrain type (flat, coastal, complex) so readers immediately see which conclusions generalize to Perdigão-like terrain.

**We thank the reviewer for offering an opportunity to make our literature review more useful to other researchers. We have introduced more site-specific characteristics throughout our literature review.**

2.    Be consistent with units formatting: use parentheses "[m s$^{-1}$]" consistently instead of mixing "[]" and "()" in figure captions. (Figure 1 and others show mixed usage.)

**We thank the reviewer for this observation. We have adjusted Figure 1 as well as other figures to reflect a consistent "[]" usage.**

3.    Move subplot descriptions from the long caption into a one-line title for each subplot (improves readability for multi-panel figures such as Figs. 3 -13). This was hard to follow when flipping between caption and panels.

**We thank the reviewer for helping improve the readability of our figures. We have adapted figures 2-15 with titles.**

4.    Line 140: double words "to to"

**Thank you. We have removed the additional "to".**

5.    Two sentences in 3.2 (lines ~394ff) are nearly identical and can be combined for concision: the point about TKE HHPI improving with height for all sites can be consolidated.

**Thank you. We adapt this line 547 to read:**

"The TKE HHPI profile consistently shows higher HHPI than the other two metrics overall, regardless of the location (Fig. 16a,b,c)."

---

## Author Comment (AC4)

**We thank both the editors and reviewers for considering this manuscript.** Below, reviewer comments are in black and **our responses are in blue bold.**

**Reviewer 3**

The paper covers an important aspect of wind meteorology: atmospheric stability. It also provides a thorough analysis of experimental data to identify trends and limitations of the different methods used to characterize stability.

The introduction provides readers with an overview of research efforts to characterize the impact of stability on turbines and wind farms. It highlights the lack of general understanding of the impact on power and wear and indicates that characterizing stability is an expensive process.

The methods section details the data sampling, equations of the indicators used (L, TKE, and $\epsilon$), how the dataset was cleaned, and which averaging windows were used. The methods section also introduces a study on the impact of the Reynolds decomposition time (1 to 60 minutes) on heat flux and friction velocity.

The conclusion helps wind farm designers determine where to locate met masts and understand the risks of taking 10-meter wind speed measurements and extrapolating them to derive an atmospheric stability assessment, which is likely inaccurate. The authors suggest using LES in complex terrain to better identify locations where it is important to place a measurement tower and to use Louvain group theory to identify these locations, as well as to identify similarities using Python code.

**We thank the reviewer for their thoughtful appraisal.**

Overall, the paper is well-structured and presented. However, it would have been helpful to include more justification for the analysis of the results. For example: "the plot indicates that ... because (explanation)" I provide a concrete example of this in my comment on L297.

**We thank the reviewer for an opportunity to be more complete in our discussion of our results. We provide more complete justifications throughout and address your specific comment where it is outlined in the end.**

The abstract could also be improved. It is unclear where the authors provide general information and where they present the paper's findings. Additionally, the analysis of the sampling window for heat flux and friction velocity, which occupies an important portion of the paper, is not mentioned.

**We thank the reviewer for their feedback on the abstract. We adapt our more general comment about the sampling window to be more direct. We also shift the sentence structure to more clearly separate specific and general claims.**

**"Characterizing atmospheric stability becomes challenging in heterogeneous complex terrain. We use data from 47 meteorological towers associated with the Perdigão field campaign to recommend data processing approaches and to assess the limitations of shorter or fewer towers. We quantify atmospheric stability according to the Obukhov Length, the turbulence kinetic energy, and the turbulence dissipation rate using two decomposition periods, including consistent 10 minute periods to match convention in the wind energy community and consistent 30 minute periods to match convention in the atmospheric science community. We also demonstrate a methodology that can indicate the necessary number and location of towers to characterize atmospheric stability. We find that the 10 minute Reynolds decomposition window underestimates turbulence patterns. Additionally, 10 m measurements do not provide reliable 100 m hub-height stability predictions. Holistically, this work addresses challenges in relying on sparse surface measurements."**

Below are some general comments and questions that I recommend addressing in the paper.

- Can you define or explain the tilt correction methodology of the sonic anemometer?

**We thank the reviewer for pointing out this oversight. The section starting on line 154 now includes a reference and discussion of the planar fit method used for this analysis:**

**"The sonic anemometer data were then tilt-corrected using the planar fit method following the approach of Wilczak et al. (2001)."**

- What about cloud cover? Is Perdigao rarely covered by clouds, or were cloudy days filtered out?

**We thank the reviewer for this question. Clouds were identified on several days throughout the field campaign and were logged on a daily basis in the supplement to Fernando et al. (2019). The revised analysis now filters out these days and the motivation for this screening is discussed, starting on line 174:**

**"[D]ata on days with clouds or precipitation were screened out of the analysis. The Perdigão field log (Supplement to Fernando et al., 2019) continues a daily breakdown of the synoptic conditions according to eight European standard patterns defined in Santos et al. (2016). This field log, along with determining the relevant synoptic**

regime for a day, also notes relevant weather patterns, such as cloud presence and precipitation. Based on this analysis, we screen out any days that include low or medium clouds as well as those with fog or precipitation. In situations where these weather patterns only affect targeted hours of the day, we remove the entire day to avoid biasing diurnal patterns. The net result of this screening is that 22 of the 45 days are removed (Appendix A)."

We also include a discussion of shading starting on line 352 with a new figure, Fig. 3:

[Figure]

Figure 3. Comparison between expected and measured incoming shortwave radiation adjacent to each of the three 100 m towers. The line represents the mean value and

the band represents the standard error. (a) tse02 (SW ridge) cloud days only; (b) tse07 (valley) cloud days only; (c) tse12 (NE ridge) cloud days only; (d) tse02 (SW ridge) cloudless days only; (e) tse07 (valley) cloudless days only; (f) tse12 cloudless days only.

"Before proceeding with stability analysis, we ensured that influences like topographic shading of our measurements did not directly impact parts of our analysis. In particular, we ensured that our designated "stable" and "unstable" hours did not experience inequitable solar exposure between towers. Importantly, these designations also translated to our CFAW-CM convergence analysis, which informed the Reynolds decomposition window.

"All three 100 m towers likely experience shading during some part of the day. The incoming shortwave solar radiation near all towers shows a typical diurnal cycle for a mid-latitude land-based site (Fig. 3). This diurnal cycle is defined by low (i.e. zero) incoming shortwave radiation in the late (18-24 UTC) and early morning (00-04 UTC) hours with an increase throughout the morning (05-11 UTC), a midday peak (12-14 UTC), and a late afternoon drop (15-17 UTC) (Fig. 3). Within this diurnal cycle, differences in solar exposure throughout the day emerge. These differences in solar exposure contribute to differences in residuals. While negligible positive residuals may reflect measurement uncertainties, negative residuals are assumed to reflect shading. Both the morning (05-07 UTC) and early evening (15-17 UTC) experience shading (Fig. 3). Cloudy days (Fig. 3a,c,e) naturally exhibit more shading than cloudless days (Fig. 3b,d,f) near each of the three 100 m towers. This diurnal difference in shading is especially pronounced in the valley, where negligible shading is suggested during cloudless days (Fig. 3d). Non-cloud-based shading due to diurnal differences in solar exposure also influences the two 100 m ridge towers. While shading in the valley is only imposed by clouds (Fig. 3c,d), the two ridges show stronger terrain-induced shading (Fig. 3a,b,e,f). Further, while the SW ridge experiences minimal terrain-induced shading in the early evening (Fig. 3b), the NE ridge shows the strongest terrain-induced shading during both the early morning and early evening (Fig. 3f). Local canopy effects may then contribute–but not-define–non-cloud shading. The NE ridge shows more diurnal shading than the SW ridge and also has a taller canopy cover. However, the valley has canopy cover but experiences no non-cloud shading. Further, the SW ridge, with a negligible canopy cover, does experience non-cloud shading. Thus, shading from the larger ridge topography may also be contributing to the experience diurnal shading. Regardless of the exact shading source, these site-based differences in shading suggest that sites do not have equivalent diurnal exposure to the incoming solar radiation. Our CFAW-CM analysis, described below, accounts for these differences in solar exposure by restricting our analysis to cloudless days and our stable and unstable periods to 00-02 UTC and 12-14 UTC, respectively."

-- Along those lines, what is the sun exposure near the ridges and in the valley? Is it full sun between 12 p.m. and 4 p.m., or is it partly shadowed by the terrain?

**We thank the reviewer for this question. To address the reviewer's curiosity, we performed an additional analysis and determined that terrain-induced shading may affect both the SW ridge and NE ridge after 15 UTC. As such, we adapt our unstable definition to include 12-14 UTC. The analysis is summarized starting on line 181:**

**"Radiometer measurements informed the second, terrain-based, shading screening. These measurements–located near the 100 m towers at tse02, tse07, and tse12–were collected at either 20 m or 30 m and reported at 5 min intervals. These (cloudless, as defined above) incoming shortwave radiation measurements were diurnally-averaged and compared to the amount of incoming shortwave radiation expected by the Ineichen-Perez clear sky model (Ineichen and Perez, 2002), as defined in the python package pvlib (Anderson et al., 2023). The residual (measured-expected) was then assumed to trace hours where terrain shading might be assumed to occur. Instead of discarding data associated with shading, by identifying inequitable patterns in solar exposure between the towers, we were able to tailor our definitions of "stable" and "unstable" to only focus on hours that were not affected by these differences. Shown later, these stable and unstable hour definitions also inform our determination of an appropriate Reynolds decomposition window for two of three stability metrics. Taken together, these two shading evaluations–albeit strict–are crucial to minimize spatial and temporal biases between towers and support a more equitable evaluation."**

**The results of this additional analysis are presented in a new Fig. 3 and discussed starting on line 352:**

[Figure]

**Figure 3. Comparison between expected and measured incoming shortwave radiation adjacent to each of the three 100 m towers. The line represents the mean value and the band represents the standard error. (a) tse02 (SW ridge) cloud days only; (b) tse07 (valley) cloud days only; (c) tse12 (NE ridge) cloud days only; (d) tse02 (SW ridge) cloudless days only; (e) tse07 (valley) cloudless days only; (f) tse12 cloudless days only.**

**"Before proceeding with stability analysis, we ensured that influences like topographic shading of our measurements did not directly impact parts of our analysis. In particular, we ensured that our designated "stable" and "unstable" hours did not experience inequitable solar exposure between towers. Importantly, these**

**designations also translated to our CFAW-CM convergence analysis, which informed the Reynolds decomposition window.**

**"All three 100 m towers likely experience shading during some part of the day. The incoming shortwave solar radiation near all towers shows a typical diurnal cycle for a mid-latitude land-based site (Fig. 3). This diurnal cycle is defined by low (i.e. zero) incoming shortwave radiation in the late (18-24 UTC) and early morning (00-04 UTC) hours with an increase throughout the morning (05-11 UTC), a midday peak (12-14 UTC), and a late afternoon drop (15-17 UTC) (Fig. 3). Within this diurnal cycle, differences in solar exposure throughout the day emerge. These differences in solar exposure contribute to differences in residuals. While negligible positive residuals may reflect measurement uncertainties, negative residuals are assumed to reflect shading. Both the morning (05-07 UTC) and early evening (15-17 UTC) experience shading (Fig. 3). Cloudy days (Fig. 3a,c,e) naturally exhibit more shading than cloudless days (Fig. 3b,d,f) near each of the three 100 m towers. This diurnal difference in shading is especially pronounced in the valley, where negligible shading is suggested during cloudless days (Fig. 3d). Non-cloud-based shading due to diurnal differences in solar exposure also influences the two 100 m ridge towers. While shading in the valley is only imposed by clouds (Fig. 3c,d), the two ridges show stronger terrain-induced shading (Fig. 3a,b,e,f). Further, while the SW ridge experiences minimal terrain-induced shading in the early evening (Fig. 3b), the NE ridge shows the strongest terrain-induced shading during both the early morning and early evening (Fig. 3f). Local canopy effects may then contribute–but not-define–non-cloud shading. The NE ridge shows more diurnal shading than the SW ridge and also has a taller canopy cover. However, the valley has canopy cover but experiences no non-cloud shading. Further, the SW ridge, with a negligible canopy cover, does experience non-cloud shading. Thus, shading from the larger ridge topography may also be contributing to the experience diurnal shading. Regardless of the exact shading source, these site-based differences in shading suggest that sites do not have equivalent diurnal exposure to the incoming solar radiation. Our CFAW-CM analysis, described below, accounts for these differences in solar exposure by restricting our analysis to cloudless days and our stable and unstable periods to 00-02 UTC and 12-14 UTC, respectively."**

Similarly, for the heat flux analysis, c**an the changes in sun exposure throughout the day be excluded?** For example, should we expect the results between 12 and 13 hours to be the same as those between 15 and 16 hours, or should we expect some variability in e.g. the sun radiation from the ground?

**We thank the reviewer for this question. To address the reviewer's curiosity, we performed an additional analysis and determined that terrain-induced shading may**

affect both the SW ridge and NE ridge after 15 UTC. As such, we adapt our unstable definition to include 12-14 UTC. The analysis is summarized starting on line 181:

"Radiometer measurements informed the second, terrain-based, shading screening. These measurements–located near the 100 m towers at tse02, tse07, and tse12–were collected at either 20 m or 30 m and reported at 5 min intervals. These (cloudless, as defined above) incoming shortwave radiation measurements were diurnally-averaged and compared to the amount of incoming shortwave radiation expected by the Ineichen-Perez clear sky model (Ineichen and Perez, 2002), as defined in the python package pvlib (Anderson et al., 2023). The residual (measured-expected) was then assumed to trace hours where terrain shading might be assumed to occur. Instead of discarding data associated with shading, by identifying inequitable patterns in solar exposure between the towers, we were able to tailor our definitions of "stable" and "unstable" to only focus on hours that were not affected by these differences. Shown later, these stable and unstable hour definitions also inform our determination of an appropriate Reynolds decomposition window for two of three stability metrics. Taken together, these two shading evaluations–albeit strict–are crucial to minimize spatial and temporal biases between towers and support a more equitable evaluation."

The results of this additional analysis are presented in a new Fig. 3 and discussed starting on line 352:

[Figure]

**Figure 3. Comparison between expected and measured incoming shortwave radiation adjacent to each of the three 100 m towers. The line represents the mean value and the band represents the standard error. (a) tse02 (SW ridge) cloud days only; (b) tse07 (valley) cloud days only; (c) tse12 (NE ridge) cloud days only; (d) tse02 (SW ridge) cloudless days only; (e) tse07 (valley) cloudless days only; (f) tse12 cloudless days only.**

**"Before proceeding with stability analysis, we ensured that influences like topographic shading of our measurements did not directly impact parts of our analysis. In particular, we ensured that our designated "stable" and "unstable" hours did not experience inequitable solar exposure between towers. Importantly, these**

designations also translated to our CFAW-CM convergence analysis, which informed the Reynolds decomposition window.

"All three 100 m towers likely experience shading during some part of the day. The incoming shortwave solar radiation near all towers shows a typical diurnal cycle for a mid-latitude land-based site (Fig. 3). This diurnal cycle is defined by low (i.e. zero) incoming shortwave radiation in the late (18-24 UTC) and early morning (00-04 UTC) hours with an increase throughout the morning (05-11 UTC), a midday peak (12-14 UTC), and a late afternoon drop (15-17 UTC) (Fig. 3). Within this diurnal cycle, differences in solar exposure throughout the day emerge. These differences in solar exposure contribute to differences in residuals. While negligible positive residuals may reflect measurement uncertainties, negative residuals are assumed to reflect shading. Both the morning (05-07 UTC) and early evening (15-17 UTC) experience shading (Fig. 3). Cloudy days (Fig. 3a,c,e) naturally exhibit more shading than cloudless days (Fig. 3b,d,f) near each of the three 100 m towers. This diurnal difference in shading is especially pronounced in the valley, where negligible shading is suggested during cloudless days (Fig. 3d). Non-cloud-based shading due to diurnal differences in solar exposure also influences the two 100 m ridge towers. While shading in the valley is only imposed by clouds (Fig. 3c,d), the two ridges show stronger terrain-induced shading (Fig. 3a,b,e,f). Further, while the SW ridge experiences minimal terrain-induced shading in the early evening (Fig. 3b), the NE ridge shows the strongest terrain-induced shading during both the early morning and early evening (Fig. 3f). Local canopy effects may then contribute–but not-define–non-cloud shading. The NE ridge shows more diurnal shading than the SW ridge and also has a taller canopy cover. However, the valley has canopy cover but experiences no non-cloud shading. Further, the SW ridge, with a negligible canopy cover, does experience non-cloud shading. Thus, shading from the larger ridge topography may also be contributing to the experience diurnal shading. Regardless of the exact shading source, these site-based differences in shading suggest that sites do not have equivalent diurnal exposure to the incoming solar radiation. Our CFAW-CM analysis, described below, accounts for these differences in solar exposure by restricting our analysis to cloudless days and our stable and unstable periods to 00-02 UTC and 12-14 UTC, respectively."

- The paper does not specify the timeframe of the data. Did you select a few days or years? Which exact dates do the data represent? During the filtering process, are significant time periods (days or weeks) flagged out? What impact does this have on, for example, the 60-minute sampling? Were dummy values inserted, or were full 60-minute blocks removed?

We thank the reviewer for this inquiry. Line 104 notes that we start with the 45-day period of 1 May - 15 June 2017. Based on the reviewer's comments above regarding clouds, we re-performed our analysis after screening out 22 of the 45 available days of the field campaign. This process is described starting on line 174:

**"[D]ata on days with clouds or precipitation were screened out of the analysis. The Perdigão field log (Supplement to Fernando et al., 2019) presents a daily breakdown of the synoptic conditions according to eight European standard patterns defined in Santos et al. (2016). This field log, along with determining the relevant synoptic regime for a day, also notes relevant weather patterns, such as cloud presence and precipitation. Based on this analysis, we screen out any days that include low or medium clouds as well as those with fog or precipitation. In situations where these weather patterns only affect targeted hours of the day, we remove the entire day to avoid biasing diurnal patterns. The net result of this screening is that 22 of the 45 days are removed (Appendix A)."**

- Regarding your use of the Louvain community detection algorithm, are any tolerances or parameters used to create the groupings? It would be useful for the reader to know what threshold was used to determine the borders between the groups.

**We thank the reviewer for this close observation. The Louvain algorithm can be tuned to several parameters, including a resolution as well as a modularity gain threshold. We stick to the default parameters in our analysis and to address this confusion in the manuscript, we now mention on line 337:**

**"While the Louvain community detection algorithm is also sensitive to several tuning parameters, we make no adjustments to the default modularity gain threshold (1e-7) or resolution (1) and introduce no artificial seeds or establish restrictions on the number of optimization cycles."**

- You mention hub height as being important in characterizing stability. On what criteria was the hub height chosen? Can stability be characterized by parameters other than turbine height, or is this choice based on the assumption that we are mostly interested in knowing what the turbine "sees"? Please explain this in the paper.

**We thank the reviewer for this inquiry. We base our criteria on the hub height of the turbine both because the hub-height is a standard measure as well as the fact that hub height aligns well with the top altitude available for the measurement towers. We clarify this decision in the text starting on line 309:**

**"While many representative heights may be potentially considered in a wind farm resource assessment, the hub height is one such representative measurement. Further, the hub height, with typical values near 100 m, naturally aligns with the highest available measurement from the meteorological towers during the Perdigão field campaign."**

Also, I have two comments related to lines:

L62: It's unclear what is meant by "stretch into an ellipse". From where is the ellipse seen? (The wake is 3D).

**Thank you, we adapt this line, now line 55, to read "stretch into a 3D ellipse."**

L297: "unstable friction velocity ogives at all tower locations show asymptotic behavior for a 30 min averaging period and a shift to mesoscale fluctuations for a 60 min averaging period"

-> Please help the reader by explaining that this is seen by the increase in u* for the 60-min average period, or otherwise if needed.

**We thank the reviewer for the opportunity to be more clear and complete in our discussion of the results, especially in our discussion of the appropriate Reynolds decomposition section.**

**We have explicitly gone through and restructured this section to be more clear by discussing these ogive-type results tower by tower. As such, we discuss the unstable friction velocities first in the valley on line 389:**

**"During unstable conditions [in the valley], heat fluxes (Fig. 4f) level off after 10 min and increase again between 30-60 min whereas friction velocities (Fig. 5f) stabilize at 5 min before increasing again after 30 min."**

**Then on line 395 for the two ridges:**

**On the SW ridge, unstable friction velocities (Fig. 5c)–especially above 10 m–stabilize by 10 min, whereas on the NE ridge they increase until about 20 min at heights above 20 m and remain constant from 1-30 min at heights at or below 20 m) (Fig. 5i).**

**Together, we hope that this makes our discussion of the unstable friction velocities more clear and complete.**